# Active Timepoint Selection for Learning Measure-Valued Trajectories

Nicolas Huynh [1]    Mihaela van der Schaar [1]

## Abstract

Inferring continuous probability paths from sparse snapshots is a fundamental challenge in domains like single-cell biology, where high-fidelity data acquisition is often destructive and constrained by prohibitive sequencing costs. This motivates the need for active learning strategies to strategically select optimal measurement times. However, designing active learning policies for this setting remains an open problem: the target objects reside on the infinite dimensional Wasserstein space where standard Euclidean metrics are ill-defined, and current interpolation methods lack epistemic uncertainty quantification. We introduce a framework which extends active experimentation to the space of measures. By leveraging Linearized Optimal Transport (LOT), we map distributional snapshots into a tangent space amenable to Gaussian Process modeling, allowing us to construct a tractable probabilistic surrogate for the underlying probability path. This yields an acquisition policy that iteratively selects measurement times to minimize uncertainty. Empirical results demonstrate that our strategy outperforms uncertainty-agnostic baselines on both synthetic and real-world datasets.

## 1. Introduction

**Measure-valued trajectories.** Inferring the temporal evolution of a probability distribution (i.e., a probability path $\{\mu_t\}_{t \in [0,1]}$) is a fundamental challenge across scientific domains, ranging from fluid dynamics (Benamou & Brenier, 2000) to macroeconomics (Achdou et al., 2022). This problem is particularly acute in single-cell biology (Wagner et al., 2016), where cellular differentiation is modeled as a dynamic process in the high-dimensional space of gene expressions (Trapnell et al., 2014). In such settings, full

continuous trajectories are often not observable. Instead, we only have limited access to destructive snapshots, which are empirical measures at discrete time points. The central task is therefore one of *distributional interpolation*: recovering the underlying continuous trajectory $t \mapsto \mu_t$ given a finite set of observed marginals at different timepoints.

**Active timepoint selection.** In practice, however, data acquisition is severely constrained by cost. For instance, in single-cell transcriptomics, generating high-fidelity snapshots entails *destructive* sampling and incurs significant expenses, i.e. often *thousands of dollars per time point* (Ziegenhain et al., 2017), which precludes dense temporal sampling. Under such budgetary limits, the timing of observations becomes critical. This motivates an *active learning* framework for measure-valued processes, designed to iteratively select the next time $t^* \in [0, 1]$ to take a measurement that best contributes to estimating the underlying probability path given the past observations [1].

**Challenges.** In this setting, active learning presents distinct challenges. First, the geometry of the output space is inherently non-Euclidean. Standard active learning methods, such as those based on Gaussian Processes (GPs) (Schulz et al., 2018; Williams & Rasmussen, 1995), assume vector-valued outputs equipped with Euclidean metrics. In contrast, probability measures live in a nonlinear space that is more naturally described by Wasserstein geometry (Ambrosio et al., 2005). Second, this makes uncertainty quantification over measures particularly challenging. Active learning for regression and classification typically relies on acquisition functions that require a notion of epistemic uncertainty, but such uncertainty is not readily available in current distribution interpolation methods (Lipman et al.; Rohbeck et al., 2025). Finally, measure-valued dynamics are often strongly non-stationary. For example, cellular development can vary dramatically in speed: extended periods of homeostasis may be punctuated by rapid, transient branching events (Haghverdi et al., 2016). As a result, uniformly spaced acquisition times can be highly suboptimal.

**Method.** We propose an active timepoint selection strategy for measure-valued trajectories that addresses the challenges above. Our key idea is to *linearize* the Wasserstein space by

---

[1]DAMTP, University of Cambridge. Correspondence to: Nicolas Huynh <nvth2@cam.ac.uk>.

*Proceedings of the 43rd International Conference on Machine Learning*, Seoul, South Korea. PMLR 306, 2026. Copyright 2026 by the author(s).

---

[1]The intended regime of our work is active acquisition with *expensive* snapshots, not real-time selection.

lifting each observed snapshot $\mu_t$ to a tangent space. Concretely, we use Linearized Optimal Transport (LOT) (Wang et al., 2013; 2025) to map $\mu_t$ to a tangent vector at a fixed reference measure. We then compress these tangent vectors into a low-dimensional representation and place a *warped* Gaussian Process (GP) prior over the resulting temporal coefficients. The GP posterior induces a practical surrogate for epistemic uncertainty, while the warping accounts for non-stationary dynamics by allowing time to be reparameterized. Finally, we leverage the GP's quantified epistemic uncertainty to determine the next time point, $t^*$, at which to take a measurement.

> **Contributions.** *Conceptually*, we formulate the problem of active learning for measure-valued trajectories, extending active experimentation to the space of measures. *Technically*, we construct a tractable probabilistic surrogate in the Wasserstein space by combining Linearized Optimal Transport with multi-output Gaussian Processes. *Empirically*, we demonstrate that our acquisition strategy outperforms uniform and random baselines on both synthetic and real-world datasets.

## 2. Related Work

**Active learning.** Active learning is well-established for scalar or categorical targets in Euclidean spaces. Early and widely used heuristics include *uncertainty sampling*, which queries points with maximal predictive ambiguity (Lewis, 1995), and *query-by-committee* (Seung et al., 1992). A complementary Bayesian perspective casts acquisition as optimal experimental design, choosing inputs that maximize expected information gain (Houlsby et al., 2011). Other works extend uncertainty and information-based acquisition to deep models using approximate Bayesian inference (e.g., MC dropout (Gal et al., 2017)). More closely related is (Singh et al., 2005), which actively select timepoints to better fit Euclidean-valued gene-expression curves. However, none of these works tackles active learning in the space of distributions, which is our focus.

**Distribution regression.** A related (but directionally reversed) line of work studies *distribution regression*, where the input is probability measures and the output lies in $\mathbb{R}^d$ (or a Hilbert space). Classical approaches embed input distributions into an RKHS via kernel mean embeddings and then perform (kernel) regression (Póczos et al., 2013; Szabó et al., 2016; Muandet et al., 2017; Law et al., 2018). In contrast, our setting treats *time* as the covariate and the *output* as a distribution. This connects more directly to *distributional interpolation* and *trajectory inference* methods that learn probability flows or stochastic processes consistent with observed marginals, including flow-matching and score-matching formulations, multi-marginal extensions,

and Schrödinger-bridge–based approaches (Lipman et al.; Tong et al., 2024; Lee et al., 2025). However, these methods are primarily *reconstruction* methods: given fixed snapshots, they typically return a single learned probability flow or fitted stochastic dynamics, and hence a single induced marginal path. By contrast, active acquisition requires *epistemic uncertainty* over plausible probability paths in order to decide which time point to measure next. Our framework models this uncertainty, making it suitable for active learning.

**Linearized Optimal Transport (LOT).** LOT embeds measures into the tangent space of a reference distribution (Wang et al., 2013; Kolouri et al., 2016). This technique has proven powerful for pattern recognition tasks on measures, such as classification and barycenter estimation, by enabling the use of linear classifiers and regression in the tangent plane (Moosmüller & Cloninger, 2023). However, none of these works focus on the active learning problem with measure-valued trajectories, which requires providing a notion of uncertainty that we introduce in this work.

## 3. Problem Setup

### 3.1. Regression in the space of distributions

**Notations.** Let $\mathcal{X} \subseteq \mathbb{R}^d$ be a feature space. We consider a probability path, i.e., a time-varying function in the space of probability measures $\mu : [0, 1] \to \mathcal{P}_2(\mathcal{X})$. For notational convenience, in what follows we denote the measure at time $t$ by $\mu_t$ instead of $\mu(t)$, and note that the maximum time can be set to an arbitrary $t_{\max}$ after scaling. $\mathcal{P}_2(\mathcal{X})$ denotes the space of probability measures on $\mathcal{X}$ with finite second moments, defined as:

$$\mathcal{P}_2(\mathcal{X}) := \left\{ \rho \in \mathcal{P}(\mathcal{X}) : \int_{\mathcal{X}} \|x\|^2 \, \mathrm{d}\rho(x) < \infty \right\} \quad (1)$$

A natural metric on this space is the 2-Wasserstein metric (Villani, 2021). For any two measures $\mu, \nu \in \mathcal{P}_2(\mathcal{X})$, the metric is defined as:

$$W_2(\mu, \nu) := \left( \inf_{\pi \in \Pi(\mu, \nu)} \int_{\mathcal{X} \times \mathcal{X}} \|x - y\|^2 \, \mathrm{d}\pi(x, y) \right)^{1/2},$$
$$(2)$$

where $\| \cdot \|$ denotes the Euclidean norm and $\Pi(\mu, \nu)$ represents the set of all joint probability measures on $\mathcal{X} \times \mathcal{X}$ with marginals $\mu$ and $\nu$.

**Goal.** We assume access to a dataset of temporal snapshots $\mathcal{D} = \{(t_i, \hat{\mu}_{t_i})\}_{i=1}^N$, where each $t_i \in [0, 1]$ is a measurement time and $\hat{\mu}_{t_i}$ is an empirical measure observed at that time (from samples drawn from the marginal $\mu_{t_i}$). Our objective is to estimate the underlying probability path $\{\mu_t\}_{t \in [0,1]}$ using the snapshots in $\mathcal{D}$.

This objective is applicable to various domains, most notably in computational biology. In this context, the feature space

$\mathcal{X}$ represents the space of gene expressions (or a latent space derived from it). Each empirical measure $\hat{\mu}_{t_i}$ corresponds to the expression profiles of $n_i$ distinct cells observed at time $t_i$, and the goal is to estimate the dynamics of the cells in expression space.

## 3.2. The active learning problem

In this work, we focus on the following question: given a fixed budget of measurements $B$, how do we select the measurement times $\{t_i\}_{i=1}^B$ in order to best estimate $\{\mu_t\}_{t \in [0,1]}$?

This means that we seek an acquisition policy $\pi$ that, given the current history $\mathcal{D}$, selects the next measurement time $t^* \in [0,1]$. We assume the measurement times are not constrained to any specific order and need not be monotonic.

> **Single-cell active sequencing.** For example, biological samples can be collected and cryopreserved at dense time intervals, forming a "bank" of potential data. Since sequencing these samples is the primary cost bottleneck, processing the entire bank is often infeasible, as the cost is typically in the order of thousands of dollars per timepoint. Instead, the active learning policy must sequentially query this bank, selecting optimal time-points to thaw and sequence to minimize trajectory uncertainty under a fixed budget.

## 3.3. Key challenges

To understand the difficulty of this active learning problem, it is useful to first recall how active learning operates in standard regression. A conventional approach is to rely on two elements: a *regressor* that predicts the output given the inputs, and a quantification of *epistemic uncertainty*. In Euclidean spaces, uncertainty can be derived from the posterior variance of a probabilistic model (e.g., Gaussian Processes) or from the empirical variance of an ensemble of predictors. The active learning policy then utilizes this uncertainty to select the next query $t^*$. However, this approach falls short in our setup, due to two fundamental challenges.

① **Non-Euclidean geometry.** Standard regression models (e.g., Gaussian Processes) rely on Euclidean operations to interpolate between observations. However, $\mathcal{P}_2(\mathcal{X})$ is not a vector space, i.e. linear combinations of measures do not lead generally to valid measures. We illustrate the limitations of naive Euclidean interpolation in Figure 1. Specifically, we consider a one-dimensional Gaussian trajectory with a time-varying mean. Each distribution is represented by its density evaluated on a fixed grid, and a standard Gaussian process is fit directly to these density vectors for interpolation. As shown in the figure, this Euclidean interpolation splits mass rather than transporting it coherently.

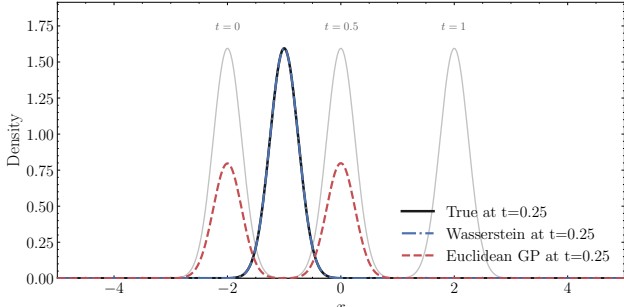

*Figure 1.* Gaussian Process regression naively applied to densities leads to poor interpolation.

While interpolation schemes compatible with the Wasserstein geometry exist, they typically operate between only *two* reference measures. An example is the displacement interpolation (McCann, 1997). Given $\mu_0, \mu_1$ in $\mathcal{P}_2(\mathcal{X})$, it is defined by:

$$\mu_t = ((1-t)\text{Id} + tT)_{\#}\mu_0, \qquad (3)$$

where $T$ is the optimal transport map from $\mu_0$ to $\mu_1$ (see Equation (4)), and $\#$ denotes the pushfoward operation. Extending this idea to settings with $N > 2$ snapshots is non-trivial. While recent works have proposed interpolation schemes for this setting (Rohbeck et al., 2025; Lee et al., 2025), they remain fundamentally deterministic and do not provide the uncertainty estimates required for active learning, which we discuss next.

② **Absence of canonical epistemic uncertainty.** Quantifying and leveraging *epistemic* uncertainty is a central idea in active learning. However, in our setting, constructing a probabilistic model over the infinite-dimensional space $\mathcal{P}_2(\mathcal{X})$ is non trivial: there is no canonical prior analogous to Gaussian Processes that respects the Wasserstein geometry while allowing for tractable posterior computation. Importantly, we note that stochastic transport formalisms do not resolve this gap. For example, multi-marginal Schrödinger Bridges define a stochastic process connecting prescribed marginals, but once the marginals are fixed and under regularity conditions, the solution is a *single* probability path. Consequently, we lack a mechanism to quantify which regions of the temporal domain show high uncertainty at the *distribution level*.

> **Desiderata.** To overcome these limitations, we seek a surrogate modeling framework that allows for probabilistic regression in $\mathcal{P}_2(\mathcal{X})$. Specifically, the model must satisfy two key criteria: (i) it must be capable of producing interpolations in $\mathcal{P}_2(\mathcal{X})$ given an arbitrary number of snapshots $N \geq 2$ and (ii) it must be *probabilistic*, providing a tractable measure of epistemic uncertainty over the trajectory to guide the acquisition policy.

# 4. Method

**Overview.** In this section, we address the challenges outlined in Section 3 and propose a framework to enable active learning on probability paths. First, to resolve the non-Euclidean and infinite-dimensional nature of $\mathcal{P}_2(\mathcal{X})$, we leverage *Linearized Optimal Transport (LOT)* (Wang et al., 2013) in order to map the observed empirical measures into a common tangent space. This effectively linearizes the Wasserstein geometry and allows us to represent distributions as vectors in this tangent space. Second, we compute a low-dimensional representation of the tangent vectors and place *Gaussian Process (GP)* priors (Rasmussen, 2003) on them. This provides a canonical notion of epistemic uncertainty, which we can use within an acquisition function to guide the sequential selection of measurement times.

## 4.1. Linearizing the Wasserstein space with LOT

The 2-Wasserstein space $(\mathcal{P}_2(\mathcal{X}), W_2)$ has a non-Euclidean geometry: measures do not interpolate naturally by linear averaging. To obtain representations of these measures, we therefore *linearize* the geometry around a fixed reference measure $\sigma \in \mathcal{P}_2(\mathcal{X})$, following the LOT framework.

**Tangent-space intuition.** Informally, the *tangent space* at $\sigma$, denoted $T_\sigma \mathcal{P}_2(\mathcal{X})$, contains the instantaneous "velocity fields" that move $\sigma$ along Wasserstein paths. One can view it as a Hilbert space of square-integrable vector fields under $\sigma$, i.e., $T_\sigma \mathcal{P}_2(\mathcal{X}) \subseteq L^2(\sigma; \mathbb{R}^d)$ (up to standard technicalities, cf. (Ambrosio et al., 2005)). Thus, our objective is to map each measure $\mu$ onto $T_\sigma \mathcal{P}_2(\mathcal{X})$.

**Optimal transport map.** To obtain this representation, we need to be able to compute the "velocity vector" from $\sigma$ to $\mu$. The geometry of $\mathcal{P}_2(\mathcal{X})$ naturally defines it via optimal transport maps. For any target measure $\mu \in \mathcal{P}_2(\mathcal{X})$, we define the optimal transport (Monge) map from $\sigma$ to $\mu$ as

$$T_{\sigma \to \mu} \in \arg\min_{T: \, T_\# \sigma = \mu} \int_{\mathcal{X}} \|x - T(x)\|^2 \, d\sigma(x). \quad (4)$$

A famous result by Brenier (1991) guarantees that if $\sigma$ is absolutely continuous with respect to the Lebesgue measure, the optimal transport map $T_{\sigma \to \mu}$ exists and is unique. The displacement field $T_{\sigma \to \mu} - \mathrm{Id}$ provides the LOT representation in the tangent space.

**LOT logarithmic map.** LOT represents a measure $\mu$ by its *displacement field* relative to a reference measure $\sigma$ through a logarithmic map. Specifically, it embeds each $\mu$ as a vector field $v_\mu \in L^2(\sigma; \mathbb{R}^d)$ defined pointwise by

$$\log_\sigma(\mu)(x) = v_\mu(x) := T_{\sigma \to \mu}(x) - x, \qquad x \in \mathcal{X}. \quad (5)$$

Under this embedding, all snapshots in $\mathcal{D}$ become elements of the same Hilbert space. Moreover, when $\mu$ and $\nu$ lie sufficiently close to $\sigma$ in Wasserstein space, squared Wasserstein distances are well-approximated by squared $L^2(\sigma)$ distances between their displacement fields, i.e. $W_2(\mu, \nu)^2 \approx \|v_\mu - v_\nu\|_{L^2(\sigma)}^2$.

This motivates choosing $\sigma$ as the Wasserstein barycenter of the observed snapshots $\{\hat{\mu}_{t_i}\}_{i=1}^N$, see Section A.1 for the definition.

**Computing the logarithmic map.** To represent each snapshot via the LOT/log map $\log_\sigma(\hat{\mu}_{t_i})$, we discretize the reference measure $\sigma$ as a weighted point cloud with $M$ landmarks, approximating it via the empirical measure $\sum_{j=1}^M w_j \delta_{z_j}$. Equivalently, it can be represented via a matrix $\mathbf{Z}_\sigma \in \mathbb{R}^{M \times d}$ and a probability weight vector $\mathbf{w} = (w_1, \ldots, w_M) \in \Delta_M$, where $\Delta_M := \{\mathbf{w} \in \mathbb{R}_+^M : \sum_{j=1}^M w_j = 1\}$. Similarly, each empirical distribution $\mu_{t_i}$ at time $t_i$ is represented as a matrix $\mathbf{Z}_i \in \mathbb{R}^{n_i \times d}$, with corresponding weight vector $\mathbf{a}^{(i)} = (a_1^{(i)}, \ldots, a_{n_i}^{(i)}) \in \Delta_{n_i}$.

For each $i \in \{1, \ldots, N\}$, we solve a discrete optimal transport problem between $(\mathbf{Z}_\sigma, \mathbf{w})$ and $(\mathbf{Z}_i, \mathbf{a}^{(i)})$, yielding a coupling $\boldsymbol{\gamma}^{(i)} \in \mathbb{R}_+^{M \times n_i}$ satisfying the marginal constraints

$$\boldsymbol{\gamma}^{(i)} \mathbf{1}_{n_i} = \mathbf{w}, \qquad (\boldsymbol{\gamma}^{(i)})^\top \mathbf{1}_M = \mathbf{a}^{(i)}.$$

The entry $\gamma_{jk}^{(i)}$ represents the amount of mass transported from the $j$-th reference landmark $z_j$ to the $k$-th target point $x_k^{(i)}$. Since discrete couplings generally allow mass splitting, a deterministic Monge map (as in Equation (4)) is not strictly defined. We therefore approximate $T_{\sigma \to \hat{\mu}_{t_i}}$ via the *barycentric projection*, which maps each reference landmark $z_j$ to the weighted barycenter of its assigned target mass:

$$\hat{T}^{(i)}(z_j) = \frac{1}{\sum_{k=1}^{n_i} \gamma_{jk}^{(i)}} \sum_{k=1}^{n_i} \gamma_{jk}^{(i)} (\mathbf{Z}_i)_k = \frac{1}{w_j} \sum_{k=1}^{n_i} \gamma_{jk}^{(i)} (\mathbf{Z}_i)_k, \quad (6)$$

where $(\mathbf{Z}_i)_k$ denotes the $k$-th row of $\mathbf{Z}_i$ and the last equality uses $\sum_k \gamma_{jk}^{(i)} = w_j$. In matrix form, the mapped landmarks are $\hat{\mathbf{Z}}_i = \mathrm{diag}(\boldsymbol{\gamma}^{(i)} \mathbf{1}_{n_i})^{-1} \boldsymbol{\gamma}^{(i)} \mathbf{Z}_i = \mathrm{diag}(\mathbf{w})^{-1} \boldsymbol{\gamma}^{(i)} \mathbf{Z}_i$

Finally, the linearized representation of the $i$-th snapshot is given by the displacement field on the reference landmarks,

$$\mathbf{V}_i = \hat{\mathbf{Z}}_i - \mathbf{Z}_\sigma \in \mathbb{R}^{M \times d}.$$

## 4.2. Low-dimensional representation

Following the LOT projection, each snapshot $\hat{\mu}_{t_i}$ is represented by a displacement matrix $\mathbf{V}_i \in \mathbb{R}^{M \times d}$. Our objective is now to obtain a low-dimensional representation from these matrices. Since LOT linearizes $\mathcal{P}_2(\mathcal{X})$ into the tangent space equipped with the $L^2(\sigma)$ geometry, we perform

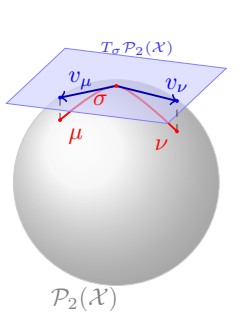

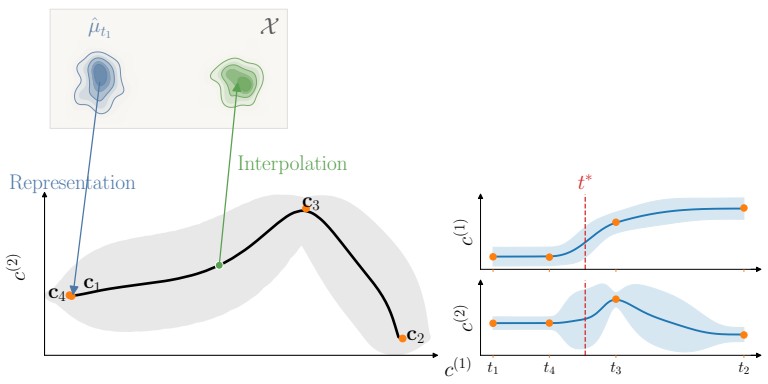

*Figure 2.* **Overview of the methodology. (Left)** Probability measures $\mu, \nu$ in Wasserstein space $\mathcal{P}_2(\mathcal{X})$ are projected onto the tangent plane $T_\sigma \mathcal{P}_2(\mathcal{X})$ via Linearized Optimal Transport (LOT). **(Right)** The active learning loop maps snapshots to latent states $c_i$ modeled by Gaussian Processes. This surrogate quantifies epistemic uncertainty to select the optimal next measurement time $t^*$.

dimensionality reduction using the corresponding discrete inner product induced by the reference weights $\mathbf{w}$. Let $\mathbf{S} := \mathrm{diag}(\sqrt{w_1}, \ldots, \sqrt{w_M}) \in \mathbb{R}^{M \times M}$ and denote the reweighted displacement field $\tilde{\mathbf{V}}_i := \mathbf{S}\,\mathbf{V}_i$. We flatten the $\tilde{\mathbf{V}}_i$ row-wise into $\tilde{\mathbf{v}}_i := \mathrm{vec}(\tilde{\mathbf{V}}_i) \in \mathbb{R}^D$, where $D = Md$, and center $\bar{\mathbf{v}} = \frac{1}{N}\sum_{i=1}^N \tilde{\mathbf{v}}_i$.

Assuming that we want to obtain a $K-$dimension representation, let $\tilde{\mathbf{U}}_K \in \mathbb{R}^{D \times K}$ be the top $K$ PCA directions of $\{\tilde{\mathbf{v}}_i - \bar{\mathbf{v}}\}_{i=1}^N$. The low-dimensional representations are then obtained as follows:

$$\mathbf{c}_i := \tilde{\mathbf{U}}_K^\top(\tilde{\mathbf{v}}_i - \bar{\mathbf{v}}) \in \mathbb{R}^K. \tag{7}$$

which yields a dataset $\tilde{\mathcal{D}} = \{(t_i, \mathbf{c}_i)\}_{i=1}^N$.

### 4.3. Modeling uncertainty with Gaussian Processes

With the low-dimensional temporal snapshots in $\tilde{\mathcal{D}}$ obtained via LOT, our objective is to construct a continuous probabilistic mapping that allows for interpolation via the observed data while quantifying epistemic uncertainty. To satisfy these two desiderata, we consider a surrogate model based on Gaussian Processes (GP).

**Priors and observation model.** We model the temporal evolution of the latent state $\mathbf{c} \in \mathbb{R}^K$ as a vector-valued function $\mathbf{f} : [0, 1] \to \mathbb{R}^K$ governed by a Multi-Output Gaussian Process (MOGP). We assume a zero-mean prior with a matrix-valued kernel $\mathbf{K}(t, t')$:

$$\mathbf{f} \sim \mathcal{GP}\left(\mathbf{0}, \mathbf{K}(\cdot, \cdot)\right), \tag{8}$$

where $\mathbf{K}(t, t') \in \mathbb{R}^{K \times K}$ is a positive semi-definite kernel matrix. The entry $[\mathbf{K}(t, t')]_{jj'}$ encodes the covariance between the $j$-th and $j'$-th latent dimensions at times $t$ and $t'$. This general formalism allows us to capture correlations between principal components if desired, though one may also

proceed with the simplifying assumption of independence, in which case $\mathbf{K}(t, t')$ is diagonal (as we do in Section 5).

We assume the observed coefficients $\mathbf{c}_i$ are noisy realizations of the latent trajectory:

$$\mathbf{c}_i = \mathbf{f}(t_i) + \boldsymbol{\epsilon}_i, \quad \boldsymbol{\epsilon}_i \sim \mathcal{N}(\mathbf{0}, \boldsymbol{\Sigma}_{\mathrm{obs}}). \tag{9}$$

Here, $\boldsymbol{\Sigma}_{\mathrm{obs}}$ captures aleatoric uncertainty arising from finite-sample approximation error in the empirical snapshots $\hat{\mu}_{t_i}$ and potential measurement noise which propagate to the LOT embeddings.

**Posterior inference.** Conditioned on the dataset $\tilde{\mathcal{D}}$, the predictive posterior distribution for the latent trajectory $\mathbf{f}$ at a query time $t$ is a multivariate Gaussian:

$$p(\mathbf{f}(t) \mid \tilde{\mathcal{D}}) = \mathcal{N}\left(\mathbf{m}(t), \mathbf{S}(t)\right). \tag{10}$$

Here, $\mathbf{m}(t) \in \mathbb{R}^K$ denotes the predicted mean vector, while the predictive covariance matrix $\mathbf{S}(t) \in \mathbb{R}^{K \times K}$ explicitly quantifies the joint epistemic uncertainty of the latent dimensions at time $t$. The diagonal elements of $\mathbf{S}(t)$ correspond to the variances of individual components, while off-diagonal elements capture their posterior correlations.

**Reconstructing distributions from the surrogate.** To obtain the predicted measure $\mu_t$ at a query time $t$, we invert the linearization pipeline. Given a latent state $\hat{\mathbf{c}}_t \in \mathbb{R}^K$ (e.g. the posterior mean $\mathbf{m}(t)$ or a sample drawn from the GP posterior), we first map it back via the inverse PCA projection $\mathbf{y}_t = \tilde{\mathbf{U}}_K \hat{\mathbf{c}}_t + \bar{\mathbf{v}}$. This vector is reshaped into $\mathbf{Y}_t \in \mathbb{R}^{M \times d}$, and the displacement field is recovered as $\hat{\mathbf{V}}_t = \mathbf{S}^{-1}\mathbf{Y}_t$. In our discrete setting, the reconstructed measure is a point cloud supported on locations $\hat{\mathbf{Z}}_t = \mathbf{Z}_\sigma + \hat{\mathbf{V}}_t$, carrying the same weights as $\sigma$.

## 4.4. Handling non-stationarity via time warping

Standard covariance kernels (e.g., RBF or Matérn) used with GPs are typically *stationary*, implying that the correlation structure depends only on the time difference, i.e., $\mathbf{K}(t, t') = \mathbf{K}_{\text{base}}(|t - t'|)$. This encodes the assumption that the rate of change of the modeled process is constant over time. However, in domains such as biology, processes can be inherently non-stationary. For example, during cellular differentiation, cells often undergo rapid transcriptional bursts followed by prolonged periods of homeostasis (Kumar et al., 2015).

**Intrinsic time parametrization.** To address this, we model the dynamics in an *intrinsic temporal domain* where the rate of distributional change is constant. We define a warping function $\Phi : [0, 1] \to \mathbb{R}^+$ that maps physical time $t$ to an intrinsic time $\tau$, representing the cumulative arclength of the trajectory on $\mathcal{P}_2(\mathcal{X})$. This is given by the integral of the metric speed $\tau = \Phi(t) := \int_0^t \|\dot{\mu}_u\|_{W_2} \, du$ where $\|\dot{\mu}_u\|_{W_2} := \lim_{h \to 0} \frac{W_2(\mu_{u+h}, \mu_u)}{|h|}$. In this warped domain, the distribution evolves at unit speed with respect to the Wasserstein metric (see Section A.2). Consequently, a stationary kernel operating on warped inputs effectively induces a non-stationary kernel on physical time.

**Approximation and inference.** Since the continuous curve is unknown, we approximate $\Phi(t)$ using the discrete empirical snapshots. Without loss of generality, we assume the timepoints are sorted before computing the warping. We compute cumulative distances on $\hat{\tau}_i = \sum_{j=2}^i W_2(\hat{\mu}_{t_{j-1}}, \hat{\mu}_{t_j})$ for $i = 2, \dots, N$ with $\hat{\tau}_1 = 0$. To obtain a continuous mapping for any candidate time $t$, we fit a monotonic cubic spline to the pairs $\{(t_i, \hat{\tau}_i)\}_{i=1}^N$.

We incorporate this warping into the GP framework (Section 4.3) by defining the kernel $\mathbf{K}$ as the composition of a base stationary kernel $\mathbf{K}_{\text{base}}$ and the mapping $\Phi$. Specifically:

$$\mathbf{K}(t, t') = \mathbf{K}_{\text{base}}\big(\Phi(t), \Phi(t')\big). \quad (11)$$

This induces a non-stationary kernel on physical time $t$ that naturally adapts its effective lengthscale to the local speed of the distributional evolution.

## 4.5. Acquisition functions

At any iteration of the active learning loop, we select the next measurement time by maximizing an acquisition function $\alpha(t; \mathcal{D})$ over a candidate pool $\mathcal{T}_{\text{pool}}$:

$$t^* = \arg \max_{t \in \mathcal{T}_{\text{pool}}} \alpha(t; \mathcal{D}). \quad (12)$$

The acquisition function encodes the criterion used to decide where the next observation is expected to be most informative. Since our non-stationarity handling is expressed in intrinsic time, all acquisition scores are evaluated through the warped location $\tau = \Phi(t)$. Let $\hat{\tau}_{\max} := \Phi(1)$ and let $\mathbf{S}(\tau) \in \mathbb{R}^{K \times K}$ denote the GP predictive covariance of the latent coefficients at intrinsic time $\tau$ given the current dataset $\mathcal{D}$.

**Example 1: point-wise uncertainty.** A simple acquisition strategy is to query where the model is locally most uncertain. This yields the score

$$\alpha_{\text{unc}}(t; \mathcal{D}) = \text{Tr}(\mathbf{S}(\Phi(t))). \quad (13)$$

This criterion favors measurement times whose corresponding intrinsic locations have high posterior epistemic uncertainty.

**Example 2: expected integrated risk reduction.** Another possibility is to query the point expected to maximally reduce uncertainty along the entire intrinsic-time domain. We quantify the current global epistemic uncertainty by

$$\mathcal{U}(\mathcal{D}) = \int_0^{\hat{\tau}_{\max}} \text{Tr}(\mathbf{S}(\tau')) \, d\tau'. \quad (14)$$

For a candidate query at physical time $t$, let $\tau = \Phi(t)$ and let $\mathbf{C}(\tau', \tau) \in \mathbb{R}^{K \times K}$ denote the posterior cross-covariance between the latent states at $\tau'$ and $\tau$. Under the GP posterior update, the covariance at $\tau'$ after observing at $\tau$ is

$$\widetilde{\mathbf{S}}(\tau') = \mathbf{S}(\tau') - \mathbf{C}(\tau', \tau)\big(\mathbf{S}(\tau) + \mathbf{\Sigma}_{\text{obs}}\big)^{-1}\mathbf{C}(\tau, \tau'). \quad (15)$$

The expected integrated risk reduction acquisition is then

$$\alpha_{\text{EIRR}}(t; \mathcal{D}) := \mathcal{U}(\mathcal{D}) - \mathcal{U}(\mathcal{D} \cup \{(t, .)\})$$
$$= \int_0^{\hat{\tau}_{\max}} \text{Tr}\bigg( \mathbf{C}(\tau', \Phi(t)) \big(\mathbf{S}(\Phi(t)) + \mathbf{\Sigma}_{\text{obs}}\big)^{-1}$$
$$\times \mathbf{C}(\tau', \Phi(t))^{\top} \bigg) d\tau'. \quad (16)$$

# 5. Experiments

In this section, we empirically evaluate the effectiveness of our active learning framework. We first assess our method on a synthetic dataset designed to mimic branching events. We then demonstrate its real-world utility using a large-scale single-cell transcriptomics dataset.

## 5.1. Experimental setup

**Baselines.** To quantify the benefits of uncertainty-aware sampling, we compare our strategy against two standard uncertainty-agnostic baselines: *Random.* Measurement times are sampled uniformly at random from the candidate pool without replacement. *Uniform.* A deterministic baseline where time points are selected on a fixed equidistant grid across the temporal domain $[0, 1]$. Results with additional baselines can be found in Section C.1.

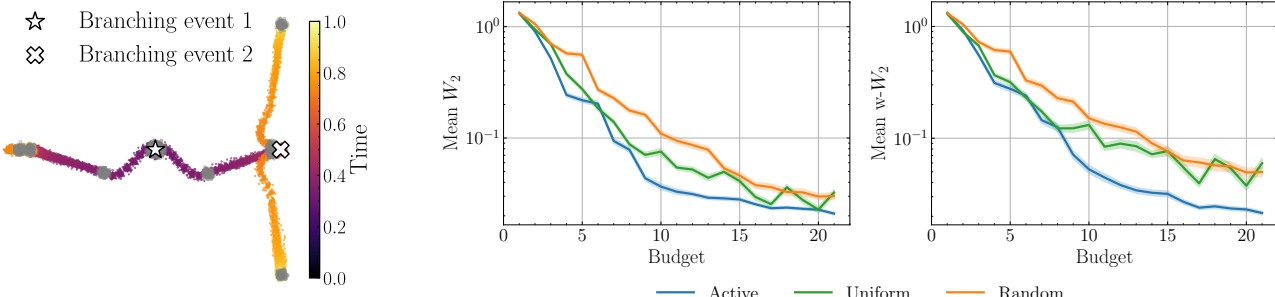

*Figure 3.* **(Left)** Visualization of the synthetic data projected in 2D. The trajectory is non-stationary with two distinct branching events (marked). **(Right)** Reconstruction performance as a function of the acquisition budget. We report the mean Wasserstein error and its velocity-weighted variant (w-$W_2$). The results are averaged over 5 seeds, and the vertical axes are presented on a logarithmic scale.

**Surrogate model.** To ensure a fair comparison, all acquisition strategies employ the identical probabilistic surrogate configurations to generate predictions and reconstruct the probability path. We use a Multi-Output Gaussian Process with independent GPs for each latent dimension, equipped with a Matérn $5/2$ kernel. Gaussian Process hyperparameters are optimized by maximizing the marginal log-likelihood (see Section B.4). Our active method uses the uncertainty-based acquisition function $\alpha_{\mathrm{unc}}$. Unless otherwise specified, all methods incorporate the intrinsic time-warping strategy described in Section 4.4 to account for non-stationary dynamics. All the methods also set the reference to be the Wasserstein barycenter of the observed distributions.

## 5.2. Synthetic experiment

**Data.** We compare the methods on a synthetic dataset designed to mimic non-stationary developmental trajectories with transient branching events. Each sample is a time series generated by simulating a low-dimensional latent process $\mathbf{z}(t) \in \mathbb{R}^{d_z}$ following an Ornstein–Uhlenbeck–type SDE with a time-dependent target mean that induces two sequential bifurcations over time windows $(a_1, b_1)$ and $(a_2, b_2)$. We also add shared oscillatory motion localized to the branching windows. Observations are obtained as $\mathbf{x}(t) = \mathbf{Q}\mathbf{z}(t)$ with $\mathbf{x}(t) \in \mathbb{R}^{10}$ and $\mathbf{Q} \in \mathbb{R}^{10 \times 2}$ having orthonormal columns. The dataset can be visualized in Figure 3, with details in Section B.3.

**Methodology.** We define an initial candidate pool of times $\mathcal{T}_{\mathrm{pool}}$ consisting of 50 time points regularly spaced in the interval $[0, 1]$. We consider varying acquisition budgets up to 21. For each method, once the budget is exhausted, we fit the surrogate model. This surrogate model is then used to obtain predicted distributions $\{\hat{\mu}_t\}_{t \in [0,1]}$ (cf. Section 4.3). We assess the quality of the inferred trajectory against the ground truth $\{\mu_t\}_{t \in [0,1]}$ on a test set $\mathcal{T}_{\mathrm{test}}$ of times. We report the Mean Wasserstein Error,

defined as $\frac{1}{|\mathcal{T}_{\mathrm{test}}|} \sum_{t \in \mathcal{T}_{\mathrm{test}}} W_2^2(\mu_t, \hat{\mu}_t)$. Additionally, to provide more granularity on the errors, we report a velocity-weighted mean Wasserstein error (w-$W_2$). This metric weighs the error at each test time point by the instantaneous speed of the ground truth process, i.e. we define w-$W_2 = \frac{1}{S} \sum_{t \in \mathcal{T}_{\mathrm{test}}} \|\dot{\mu}_t\| \cdot W_2^2(\mu_t, \hat{\mu}_t)$ where $\|\dot{\mu}_t\|$ represents a metric speed of the true probability path at time $t$ and $S = \sum_{t \in \mathcal{T}_{\mathrm{test}}} \|\dot{\mu}_t\|$ is the normalization constant. This metric therefore puts more weight on test times where the dynamics are high.

**Results.** We report these metrics in Figure 3, as a function of the budget, for 5 seeds. Our *Active* strategy consistently outperforms both the *Uniform* and *Random* baselines across the acquisition budgets. Crucially, the performance gap widens after a budget of 5, suggesting that our active learning method starts exploiting by querying difficult regions. This is also supported by the velocity-weighted metrics, confirming that the acquisition function successfully identifies and targets the non-stationary regions of the trajectory, specifically the rapid branching events, where the interpolation error is naturally highest. While the *Uniform* baseline improves stepwise as the grid density increases, the active approach provides a more sample-efficient strategy for low to moderate budgets.

**Qualitative analysis.** To investigate the mechanism driving this efficiency, we visualize the temporal allocation of measurements in Figure 4. The top panel displays the instantaneous metric speed $\|\dot{\mu}_t\|$ of the ground truth trajectory, revealing a highly non-stationary process with sharp peaks corresponding to the sequential branching events described in Section B.3. The bottom panel illustrates the specific time points selected by each strategy. We see that the *Active* strategy exhibits strong adaptivity. It concentrates its later acquisitions in the high-velocity windows (approximately $t \in [0.3, 0.4]$ and $t \in [0.7, 0.8]$).

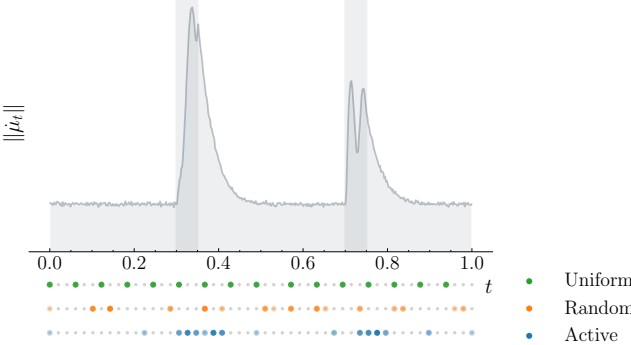

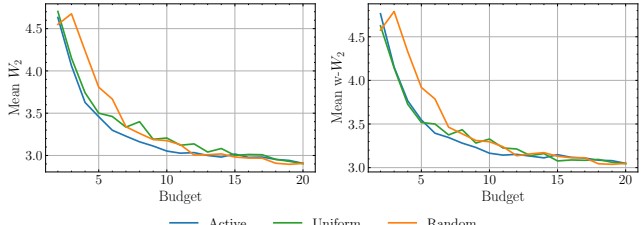

*Figure 5.* Evaluation on the single-cell reprogramming dataset from (Schiebinger et al., 2019)

*Figure 4.* Visualization of adaptive timepoint selection. **(Top)** The instantaneous metric speed $\|\dot{\mu}_t\|$ of the ground truth trajectory. **(Bottom)** Comparison of selected measurement times for a budget of 16. Darker points indicate later acquisitions.

### 5.3. Application to real-world datasets

**Data.** We evaluate our method on the large-scale single-cell RNA sequencing dataset from Schiebinger et al. (2019), which tracks the reprogramming of mouse fibroblasts into induced pluripotent stem cells (iPSCs). We specifically focus on the serum culture subset, which exhibits non-stationary developmental dynamics over an 18-day period. The dataset contains 39 distinct measurement times within the interval $t \in [0, 18]$. We project the gene expressions into a 20-dimensional latent space with PCA, and we whiten the components. We partition the snapshot timepoints into two interleaved disjoint sets, one serving as the candidate pool for timepoint selection and the other as a test set.

**Results.** We report the results in Figure 5. The active strategy achieves the lowest reconstruction error in the low-to-moderate budget regime ($B \leq 12$) where uniform grids are most likely to miss brief, rapidly changing phases of the underlying reprogramming dynamics. As expected, as we increase the acquisition budget past this point, the performance gap between methods shrinks: once the schedule becomes dense, even uniform or random sampling is likely to cover most transient phases, reducing the advantage of active selection. To summarize, the clearest advantage of our method is in the *high-precision, moderate-budget regime,* while the differences are smaller in the low-precision regime and at very low or very high budgets. Finally, we also report the computational cost of our method in Section C.3.

**Sensitivity analysis.** Building on this observation, we hypothesize that our active learning method is most effective in scenarios characterized by localized heterogeneity in metric speed.

To further validate our findings, we perform a sensitivity analysis by varying the duration of the branching events. While the baseline configuration utilized a duration of $0.05$ (corresponding to the interval $[0.3, 0.35]$), we also evaluate wider intervals with durations of $0.10$ and $0.20$. For each configuration, we compute the average relative improvement of the active method over the uniform and random baselines, aggregated across all budgets and 5 seeds. Results in Table 1 show that the Active strategy yields the most significant gains when branching events are highly localized (length $0.05$). This advantage persists at $0.10$ but diminishes as windows widen to $0.20$. At this width, improvement over the Uniform baseline becomes negative for the mean Wasserstein error; however, the velocity-weighted metric gap remains positive, indicating the method still effectively targets high-velocity regions. These results confirm that the strength of our active approach lies in its ability to precisely target sharp, transient dynamics that uniform sampling intervals are likely to miss.

**Additional dataset.** In Section C.4, we perform the same experiment on a labor market dataset from (Flood et al., 2024), where we show that our method preferentially identifies and explores periods where the distribution changes most rapidly (around the onset of the COVID-19 pandemic).

### 5.4. Ablations

We ablate four components of our surrogate/acquisition pipeline: (i) replace the default Matérn-$5/2$ GP kernel with an RBF kernel; (ii) fix the LOT reference $\sigma$ to the initially observed snapshot (no Wasserstein-barycenter refitting at each iteration); (iii) reduce the PCA basis rank to $K = 2$; and (iv) disable intrinsic-time warping.

*Table 1.* Average relative improvement of the active method compared to the baselines across 5 seeds. $l$ controls the duration of the branching events. Subscripts denote 90% CI.

| | vs. Uniform | | vs. Random | |
|---|---|---|---|---|
| $l$ | Rel. $W_2$ | Rel. w-$W_2$ | Rel. $W_2$ | Rel. w-$W_2$ |
| 0.05 | $0.231_{(0.009)}$ | $0.357_{(0.011)}$ | $0.342_{(0.122)}$ | $0.454_{(0.094)}$ |
| 0.10 | $0.189_{(0.005)}$ | $0.277_{(0.004)}$ | $0.397_{(0.104)}$ | $0.464_{(0.077)}$ |
| 0.20 | $-0.172_{(0.011)}$ | $0.071_{(0.005)}$ | $0.118_{(0.080)}$ | $0.295_{(0.057)}$ |

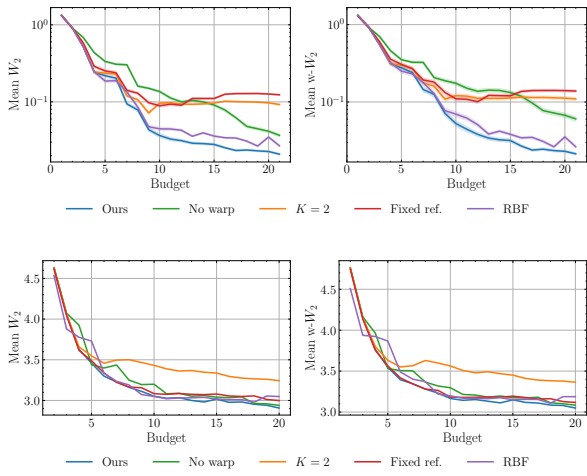

*Figure 6.* **Ablation study.** We evaluate the impact of: (i) replacing the Matérn-5/2 kernel with an **RBF** kernel; (ii) fixing the LOT reference $\sigma$; (iii) reducing the PCA basis rank to $K = 2$; and (iv) disabling the **intrinsic time warping** strategy (*No warp*).

**Results.** We report the results for both the synthetic and the fibroblast reprogramming dataset in Figure 6. We observe that reducing the latent dimension to $K = 2$ results in the most significant performance drop, confirming that a sufficiently high-dimensional latent representation is critical for capturing complex transcriptomic variations. Fixing the reference distribution $\sigma$ to the initial snapshot leads to considerably higher error compared to the full method on the synthetic dataset, demonstrating the importance of updating reference to minimize the linearization error. We provide a detailed analysis on the role of the reference in Section C.2. Disabling the intrinsic time warping degrades performance, particularly in the low-budget regime, which validates the utility of our non-stationary modeling approach. Finally, the results with the RBF kernel show that different priors (injected via the kernel) can be used with our method.

## 6. Discussion

We presented a framework for active learning on measure-valued trajectories that leverages Linearized Optimal Transport and Gaussian Processes to quantify uncertainty in Wasserstein space. Our experiments demonstrate that this strategy outperforms uncertainty-agnostic baselines by targeting high-velocity regions, such as branching events in cellular differentiation. A primary limitation of our approach is the reliance on the tangent space approximation. While iteratively updating the reference measure mitigates this, large distributional shifts may still induce linearization errors. Using multiple charts and atlas-like construction with tools from Riemannian geometry is an interesting direction for future work. Additionally, the main computational cost of our framework comes from the OT subroutines. In the

intended application regime of this paper, i.e. active acquisition settings with expensive snapshots and up to roughly $10^5$ samples per snapshot, this overhead is practical. We do not claim that million-point snapshots are fully solved: in that regime, both OT computation and memory become major constraints, and additional scalable OT approximations would be required. Finally, future work could focus on incorporating multidimensional inputs in addition to time variables.

## Acknowledgments

We thank the four anonymous ICML reviewers for their comments and suggestions. NH thanks Illumina for their funding and support.

## Impact Statement

This paper presents work whose goal is to advance the field of Machine Learning. There are many potential societal consequences of our work, none which we feel must be specifically highlighted here. Code to reproduce the experiments can be found at https://github.com/nicolashuynh/active_wass, and at the wider lab repository https://github.com/vanderschaarlab/active_wass.

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

# A. Theory

## A.1. Wasserstein barycenter

In our framework, the reference measure $\sigma$ defines the tangent space onto which all snapshots are projected. To minimize the global linearization error, we define $\sigma$ as the *Wasserstein barycenter* of the observed snapshots $\{\hat{\mu}_{t_i}\}_{i=1}^N$.

Given the snapshots and a set of weights $\{\lambda_i\}_{i=1}^N$ such that $\sum \lambda_i = 1$ (typically $\lambda_i = \frac{1}{N}$), the Wasserstein barycenter is the solution to the following optimization problem:

$$\sigma \in \arg\min_{\nu \in \mathcal{P}_2(\mathcal{X})} \sum_{i=1}^N \lambda_i W_2^2(\nu, \hat{\mu}_{t_i}). \tag{17}$$

In the discrete setting where snapshots are empirical point clouds, the barycenter $\sigma$ is also a discrete measure.

## A.2. Intrinsic time parametrization

In this section, we provide the formal derivation showing that the intrinsic time parametrization defined in Section 4.4 induces a unit-speed trajectory in the 2-Wasserstein space $\mathcal{P}_2(\mathcal{X})$.

**Setup.** Let $\mu : [0,1] \to \mathcal{P}_2(\mathcal{X})$ be an absolutely continuous curve. The metric derivative of $\mu$ at time $t$ is defined as:

$$v(t) := \|\dot{\mu}_t\|_{W_2} = \lim_{h \to 0} \frac{W_2(\mu_{t+h}, \mu_t)}{|h|}. \tag{18}$$

This limit exists for almost every $t \in [0,1]$ (Ambrosio et al., 2005). We assume the curve is regular, such that $v(t) > 0$ almost everywhere.

The intrinsic time (arc-length) warping function $\Phi : [0,1] \to [0,L]$ is defined as:

$$\tau = \Phi(t) := \int_0^t v(u)\, du, \tag{19}$$

where $L = \int_0^1 v(u)du$ is the total length of the curve. Since $v(t) > 0$, $\Phi$ is strictly increasing and absolutely continuous, admitting a well-defined inverse $\Psi := \Phi^{-1} : [0,L] \to [0,1]$.

**Reparametrization.** We define the reparametrized curve $\nu : [0,L] \to \mathcal{P}_2(\mathcal{X})$ in the warped domain as $\nu_\tau := \mu_{\Psi(\tau)}$. We aim to show that $\|\dot{\nu}_\tau\|_{W_2} = 1$ for almost every $\tau$.

*Proof.* Let $\tau \in [0,L]$ be a point where $\Psi$ is differentiable and the metric derivative of $\mu$ exists at $t = \Psi(\tau)$. Consider the metric derivative definition for $\nu$ at $\tau$:

$$\|\dot{\nu}_\tau\|_{W_2} = \lim_{k \to 0} \frac{W_2(\nu_{\tau+k}, \nu_\tau)}{|k|} = \lim_{k \to 0} \frac{W_2(\mu_{\Psi(\tau+k)}, \mu_{\Psi(\tau)})}{|k|}. \tag{20}$$

Since $\Psi$ is strictly increasing, $\Psi(\tau + k) \neq \Psi(\tau)$ for $k \neq 0$. We have:

$$\|\dot{\nu}_\tau\|_{W_2} = \lim_{k \to 0} \left( \frac{W_2(\mu_{\Psi(\tau+k)}, \mu_{\Psi(\tau)})}{|\Psi(\tau+k) - \Psi(\tau)|} \cdot \frac{|\Psi(\tau+k) - \Psi(\tau)|}{|k|} \right). \tag{21}$$

Let $\Delta t_k = \Psi(\tau+k) - \Psi(\tau)$. As $k \to 0$, it follows that $\Delta t_k \to 0$. We have:

$$\lim_{k \to 0} \frac{W_2(\mu_{t+\Delta t_k}, \mu_t)}{|\Delta t_k|} = \|\dot{\mu}_t\|_{W_2} = v(t). \tag{22}$$

Using the derivative of the inverse function $\Psi'(\tau) = 1/\Phi'(\Psi(\tau))$:

$$\lim_{k \to 0} \frac{|\Psi(\tau+k) - \Psi(\tau)|}{|k|} = |\Psi'(\tau)| = \frac{1}{|v(t)|}. \tag{23}$$

---

**Algorithm 1** Active Wasserstein Trajectory Learning

---

1: **Input:** Initial snapshots $\mathcal{D} = \{(t_i, \hat{\mu}_{t_i})\}_{i=1}^{N}$, Candidate pool $\mathcal{T}_{\text{pool}}$, Acquisition budget $B$, Latent dimension $K$.
2: **Hyperparameters:** GP kernel $K_{\text{base}}$, Reference landmark count $M$.
3: **while** budget $B > 0$ **do**
4:     // **1. Geometry linearization (Sec. 4.1)**
5:     Update reference $\sigma$ as the Wasserstein barycenter of $\mathcal{D}$.
6:     Represent $\sigma$ via landmarks $\mathbf{Z}_\sigma \in \mathbb{R}^{M \times d}$ and weights $\mathbf{w}$.
7:     **for** each $(t_i, \hat{\mu}_{t_i}) \in \mathcal{D}$ **do**
8:         Compute OT coupling $\boldsymbol{\gamma}^{(i)}$ between $(\mathbf{Z}_\sigma, \mathbf{w})$ and $\hat{\mu}_{t_i}$.
9:         Compute barycentric projection $\hat{\mathbf{Z}}_i = \text{diag}(\mathbf{w})^{-1}\boldsymbol{\gamma}^{(i)}\mathbf{Z}_i$ (Eq. 6).
10:        Compute displacement field $\mathbf{V}_i \leftarrow \hat{\mathbf{Z}}_i - \mathbf{Z}_\sigma$.
11:     **end for**
12:     // **2. Low-dimensional representation (Sec. 4.2)**
13:     Reweight and flatten: $\tilde{\mathbf{v}}_i = \text{vec}(\mathbf{S}\mathbf{V}_i)$ where $\mathbf{S} = \text{diag}(\sqrt{\mathbf{w}})$.
14:     Compute PCA basis $\tilde{\mathbf{U}}_K$ on centered vectors $\{\tilde{\mathbf{v}}_i - \bar{\mathbf{v}}\}_{i=1}^{N}$.
15:     Project snapshots to latent space: $\mathbf{c}_i \leftarrow \tilde{\mathbf{U}}_K^{\top}(\tilde{\mathbf{v}}_i - \bar{\mathbf{v}})$ (Eq. 7).
16:     // **3. Intrinsic time warping (Sec. 4.4)**
17:     Compute cumulative transport distances $\hat{\tau}_i = \sum_{j=1}^{i} W_2(\hat{\mu}_{t_{j-1}}, \hat{\mu}_{t_j})$.
18:     Fit monotone cubic spline $\Phi(t)$ to pairs $\{(t_i, \hat{\tau}_i)\}_{i=1}^{N}$.
19:     // **4. Probabilistic Modeling (Sec. 4.3)**
20:     Define warped kernel $\mathbf{K}(t, t') = \mathbf{K}_{\text{base}}(\Phi(t), \Phi(t'))$.
21:     Fit Multi-Output GP on $\tilde{\mathcal{D}} = \{(t_i, \mathbf{c}_i)\}$ to obtain posterior $p(\mathbf{f}|\tilde{\mathcal{D}})$.
22:     // **5. Acquisition (Sec. 4.5)**
23:     Select next timepoint by maximizing acquisition function:
24:     $t^* \leftarrow \arg\max_{t \in \mathcal{T}_{\text{pool}}} \alpha(t; \mathcal{D})$     (Eq. 12).
25:     // **6. Update**
26:     Acquire measurement $\hat{\mu}_{t^*}$ (perform experiment/sequence).
27:     $\mathcal{D} \leftarrow \mathcal{D} \cup \{(t^*, \hat{\mu}_{t^*})\}$.
28:     $N \leftarrow N + 1, \quad B \leftarrow B - 1$.
29: **end while**
30: **Output:** Fitted surrogate model $\mathbf{f}$ and reference $\sigma$.

---

Since $v(t) > 0$, combining these results yields:

$$\|\dot{\nu}_\tau\|_{W_2} = v(t) \cdot \frac{1}{v(t)} = 1. \tag{24}$$

Thus, the distribution evolves at unit speed with respect to the Wasserstein metric in the warped domain. $\qquad\square$

## B. Experimental Details

Code to reproduce the experiments can be found at https://github.com/nicolashuynh/active_wass, and at the wider lab repository https://github.com/vanderschaarlab/active_wass. All the experiments were conducted on a single machine equipped with a 18-Core Intel Core i9-10980XE CPU.

### B.1. Algorithm

We provide an algorithm section detailing our method in Algorithm 1.

### B.2. Computational complexity

The computational cost of our framework is primarily driven by the optimal transport computations and the Gaussian Process inference. We analyze the complexity with respect to the number of observed snapshots $N$, the average number of points per snapshot $n$, the number of reference landmarks $M$, and the ambient dimension $d$.

**LOT Embedding and Warping.** Mapping the $N$ snapshots to the tangent space requires solving $N$ discrete optimal transport problems between the reference $\sigma$ and each $\hat{\mu}_{t_i}$. Additionally, the time warping approximation requires solving $N$ pairwise OT problems between consecutive snapshots. Let $\mathcal{C}_{\mathrm{OT}}(M, n)$ denote the cost of a single transport solve (e.g., $\mathcal{O}(Mn)$ for Sinkhorn approximations or $\tilde{\mathcal{O}}((M + n)^3)$ for exact solvers). The total cost for embedding and warping is $\mathcal{O}(N \cdot \mathcal{C}_{\mathrm{OT}}(M, n))$. The subsequent barycentric projection involves matrix multiplications with complexity $\mathcal{O}(NMnd)$.

**Surrogate Construction.** Dimensionality reduction via PCA on the displacement vectors (matrix size $N \times Md$) typically incurs a cost of $\mathcal{O}(N^2 Md)$, assuming $N \ll Md$. For the probabilistic model, training $K$ independent GPs on $N$ time points is dominated by the Cholesky decomposition of the kernel matrices, scaling as $\mathcal{O}(KN^3)$.

**Active Loop.** In the active learning setting, adding a new measurement requires solving one additional OT problem and updating the GPs. Evaluating the acquisition function $\alpha(t)$ across a dense grid of $T_{\mathrm{grid}}$ candidate times requires computing predictive variances, scaling as $\mathcal{O}(T_{\mathrm{grid}}KN^2)$. Since our framework is designed for settings where data is sparse (small $N$) but high-dimensional, the cubic scaling of the GP is negligible compared to the cost of optimal transport, which constitutes the main computational bottleneck, and which can be reduced by using entropic formulations.

## B.3. Datasets.

### B.3.1. OSCILLATORY SEQUENTIAL BRANCHING DATASET

**Latent dynamics.** We generate a branching trajectory in a low-dimensional latent space $\mathbf{z}(t) \in \mathbb{R}^{d_z}$ over $t \in [t_{\min}, t_{\max}]$ using an Ornstein–Uhlenbeck–type SDE with a time-dependent target mean:

$$d\mathbf{z}(t) = -\kappa\big(\mathbf{z}(t) - \boldsymbol{\mu}(t; s)\big)\, dt + \sigma_{\mathrm{diff}}\, d\mathbf{W}_t, \tag{25}$$

where $\kappa > 0$ is the drift strength, $\sigma_{\mathrm{diff}}$ is the diffusion scale, and $\mathbf{W}_t$ is standard Brownian motion. Each trajectory is assigned binary fate labels $s = (\sigma_1, \sigma_2) \in \{\pm 1\}^2$.

**Branching schedule and target mean.** Two sequential branching events occur within time windows $(a_1, b_1)$ and $(a_2, b_2)$, with smooth gates

$$\alpha_j(t) = \mathrm{clip}\left(\frac{t - a_j}{b_j - a_j}, 0, 1\right), \quad j \in \{1, 2\}. \tag{26}$$

Define $g = \mathbb{I}\{\sigma_1 = \texttt{split\_branch\_sign}\}$ to enforce that the second split only affects one side of the first branch. The base mean in $d_z = 2$ is

$$\boldsymbol{\mu}_0(t; s) = \begin{bmatrix} g\,\beta\,\sigma_2\,\alpha_2(t) \\ \beta\,\sigma_1\,\alpha_1(t) \end{bmatrix}. \tag{27}$$

To introduce oscillatory nuisance motion localized near the branching events, we add a time-varying perturbation:

$$e_j(t) = 20\,\alpha_j(t)\big(1 - \alpha_j(t)\big), \qquad w(t) = \sin(\omega t), \tag{28}$$

$$\boldsymbol{\mu}(t; s) = \begin{bmatrix} \mu_{0,x}(t; s) + A\,e_1(t)\,w(t) \\ \mu_{0,y}(t; s) + A\,e_2(t)\,w(t) \end{bmatrix}, \tag{29}$$

where $A$ is the nuisance amplitude and $\omega$ the nuisance frequency. This adds shared oscillatory motion during the transition windows without changing the branching topology.

**Initialization.** At $t = t_{\min}$ we initialize

$$\mathbf{z}(t_{\min}) \sim \mathcal{N}(\mathbf{0},\, \sigma_{\mathrm{init}}^2 \mathbf{I}), \tag{30}$$

with independent samples across trajectories.

**Observation model.** Latent states are mapped into $\mathbf{x}(t) \in \mathbb{R}^{d_x}$ via a fixed orthonormal embedding:

$$\mathbf{x}(t) = \mathbf{Q}\mathbf{z}(t) + \boldsymbol{\epsilon}, \tag{31}$$

where $\mathbf{Q} \in \mathbb{R}^{d_x \times d_z}$ has orthonormal columns.

**Discretization.** Trajectories are simulated using Euler–Maruyama with step $\Delta t$; we record observations on the same grid.

*Table 2.* Data hyperparameters for the oscillatory sequential branching dataset

| Parameter Description | Value / Setting |
|---|---|
| Time interval | $t_{\min} = 0.0$, $t_{\max} = 1.0$ |
| Internal integration step | $\Delta t = 0.005$ |
| Latent dimension | $d_z = 2$ |
| Observed dimension | $d_x = 10$ |
| Drift strength | $\kappa = 20.0$ |
| Diffusion scale | $\sigma_{\text{diff}} = 0.2$ |
| Initial latent std | $\sigma_{\text{init}} = 0.05$ |
| Branch scale | $\beta = 2.0$ |
| Branching windows | $(a_1, b_1) = (0.30, 0.35)$, $(a_2, b_2) = (0.70, 0.75)$ |
| Split branch sign | $-1$ |
| Nuisance frequency | $\omega = 30.0$ |
| Nuisance amplitude | $A = 0.5$ |

**Data hyperparameters**  We provide the hyperparameters used to generate the data in Table 2.

### B.3.2. SCHIEBINGER REPROGRAMMING DATASET (WADDINGTON-OT)

**Source and biological context.**  We use the time-resolved single-cell RNA-seq dataset introduced in (Schiebinger et al., 2019) (mouse fibroblast reprogramming under OSKM induction). The full study contains multiple culture conditions. In our experiments we restrict to the *serum* subset. Cells are annotated with a numeric day post-induction, and two experimental batches are provided.

**Representation and preprocessing.**  We treat each time point as an empirical measure over cell embeddings. Let $\mathbf{x}_i(t) \in \mathbb{R}^{d_x}$ denote the embedding of cell $i$ at day $t$. We perform PCA to obtain $d_x = 20$ components and we whiten these components in our experiments.

**Train/eval split and candidate times.**  We use one batch, and we split the available time points into training and evaluation sets by taking a fixed fraction of times for evaluation $(0.5)$. For sampling at a requested time $t$, we select the nearest available time point within a small tolerance.

**Time points.**  Time is measured in days post-induction. The serum subset includes discrete time points spanning $t \in [0, 18]$ with half-day spacing and a few quarter-day measurements (e.g., 8.25, 8.5, 8.75, 9.5, 10.5, 11.5, 12.5, 13.5, 14.5, 15.5, 16.5, 17.5).

*Table 3.* Data hyperparameters for the Schiebinger (Waddington-OT) dataset

| Parameter Description | Value / Setting |
|---|---|
| Dataset subset | Serum |
| Batch split | Train batch index 0, test batch index 0 |
| PCA dimension | $d_x = 20$ |

### B.4. Hyperparameters.

In Section 5, all active, uniform, and random comparisons use the same linearized Wasserstein Gaussian-process surrogate. We use independent scalar GPs for the tangent-space coefficients, with a Matérn-5/2 covariance kernel and a Wasserstein arc-length time warp. The warped time coordinate is then linearly rescaled to $[0, 1]$, and output coefficients are scaled by their empirical standard deviation before fitting. The tangent PCA rank is set to 4 for the synthetic branching experiment and 20 for the single-cell experiment. The nominal Matérn lengthscale is $0.4$ in the synthetic experiment and $1.5$ in the single-cell experiment. When at least two observations are available, this value is reinitialized to the median pairwise distance between the observed, warped, rescaled time points. Kernel output scales are initialized from the empirical coefficient variance. GP

hyperparameters are optimized by maximizing the exact marginal likelihood with an L-BFGS-style SciPy optimizer, using Cholesky jitter $10^{-5}$. We place a data-scaled Gamma prior on the noise variance, with prior mean equal to $0.1$ times the empirical residual variance. The likelihood noise is initialized from the residual variance with scale $10^{-3}$ for the synthetic experiment and $10^{-2}$ for the single-cell experiment. For the synthetic experiment, the reference $\sigma$ has 512 support points, and for the single-cell experiment, it has 1024 support points.

## C. Additional Results

### C.1. Additional baselines

We consider the following additional baselines:

- **Moments embeddings**: we represent any distribution $\mu$ on $\mathbb{R}^d$ by its mean $m_\mu$ and covariance $\Sigma_\mu$, and encode it as $\psi(\mu) = (m_\mu, \mathrm{uptri}(\Sigma_\mu))$ before PCA, where $\mathrm{uptri}$ returns the upper triangular coefficients including the diagonal. At reconstruction time, we decode $\hat{\psi}$ into $(\hat{m}, \hat{\Sigma})$, project $\hat{\Sigma}$ to be PSD, and map this to the Gaussian $\mathcal{N}(\hat{m}, \hat{\Sigma})$.

- **Kernel mean embeddings**: given a distribution $\mu$, we fix anchors $Z = \{z_\ell\}_{\ell=1}^L$, and, with $k_\eta(x, z) = \exp(-\|x - z\|^2/(2\eta^2))$, we define $\psi_\mu(\ell) = \mathbb{E}_{x \sim \mu}[k_\eta(x, z_\ell)]$. At reconstruction time, from $\hat{\psi}$ we recover anchor weights $p$ by optimizing $\min_{p \in \Delta^L} \|K_{ZZ}p - \hat{\psi}\|_2^2 + \lambda\|p\|_2^2$ where $(K_{ZZ})_{\ell m} = k_\eta(z_\ell, z_m)$, which yields $\hat{\mu} = \sum_{\ell=1}^L p_\ell\, \delta_{z_\ell}$.

- **Interval midpoints**: at each acquisition step, we select the candidate time closest to the midpoint of the largest interval between adjacent observed times (both with and without time warping).

- **Spline uncertainty**: we adapt the acquisition function in (Singh et al., 2005). Note that (Singh et al., 2005) tackles a different problem: selecting time points given measurements from $k$ functions, while we focus on the active learning problem in the space of *distributions*. We adapt it to our problem by using their spline uncertainty acquisition function to our LOT+PCA coefficients (both with and without time warping).

**Results.** We report the results in Figure 7, where 1) the alternative embeddings yield drastically worse results than with the LOT embeddings, showing the importance of capturing the Wasserstein geometry with OT and 2) our acquisition function, which captures epistemic uncertainty, outperforms the other baselines.

### C.2. Distortion analysis

**Role of the reference.** The quality of the surrogate depends on how well LOT preserves Wasserstein geometry. To reduce the distortion, we choose the reference measure $\sigma$ as the Wasserstein barycenter of the observed snapshots, rather than fixing an arbitrary reference. To provide more intuition on the importance of this choice, we perform an analysis quantifying the distortion induced by $\sigma$: for pairs $\mu_i, \mu_j$, we compare the true $W_2(\mu_i, \mu_j)^2$ with its LOT approximation $\|z_{i,\sigma} - z_{j,\sigma}\|_2^2$. We report mean absolute and relative errors, as well as correlations in Figure 8, Table 4, Table 5, and show that using the Wasserstein barycenter as reference yields substantially lower distortion than using the first snapshot as reference.

*Table 4.* Linearization errors for the synthetic dataset.

| Reference $\sigma$ | Mean abs. error | Mean rel. error | Corr |
|---|---|---|---|
| Wasserstein barycenter | 0.0004 | 0.0347 | 1.000 |
| First snapshot | 0.2867 | 4.4932 | 0.966 |

*Table 5.* Linearization errors for the single-cell dataset.

| Reference $\sigma$ | Mean abs. error | Mean rel. error | Corr |
|---|---|---|---|
| Wasserstein barycenter | 1.2057 | 0.0543 | 0.9924 |
| First snapshot | 2.0529 | 0.0921 | 0.9594 |

### C.3. Runtime summary

We report the runtime summary for the single-cell experiment in Table 6.

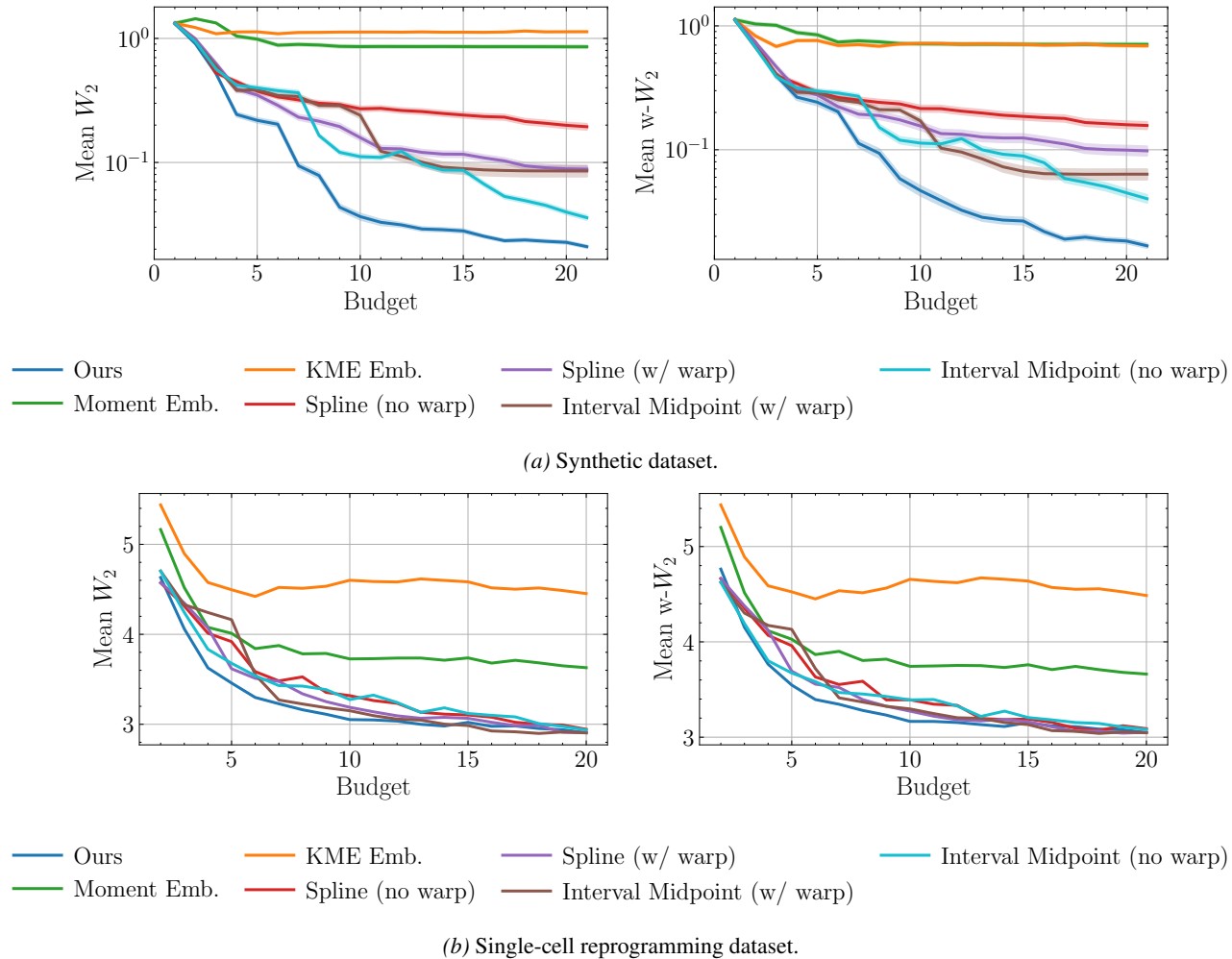

*(a)* Synthetic dataset.

*(b)* Single-cell reprogramming dataset.

*Figure 7.* Reconstruction performance as a function of the acquisition budget. We report the mean Wasserstein error and its velocity-weighted variant (w-$W_2$).

### C.4. Labor market microdata

**Data.** We conduct an experiment using real-world IPUMS-CPS monthly microdata (Flood et al., 2024). Specifically, we use monthly non-ASEC U.S. CPS samples from January 2015 to December 2021 and construct a time-indexed sequence of distributions over weekly earnings. For each individual record, we retain observations satisfying `AGE` $\geq 16$, `EARNWT` $> 0$, and `EARNWEEK` $> 0$. We represent each month as a weighted empirical measure over $\log(\texttt{EARNWEEK})$, with weights normalized within each month using `EARNWT`.

**Results.** As shown in Figure 9a, the active strategy performs similarly to the uniform baseline when error is averaged uniformly over calendar time (left panel), and yields a clear improvement under the intrinsic-time (velocity-weighted metric on the right panel). This difference is clear in Figure 9b: our method preferentially identifies and explores periods where the distribution changes most rapidly. That occurs around the onset of the COVID-19 pandemic, especially in March-April 2020. During this period, the distribution of weekly earnings shifted quickly, because there were many fewer low-earning workers in the data. This shows that our method is a general tool for adaptive experimentation on measure-valued dynamical systems, and is not restricted to biology.

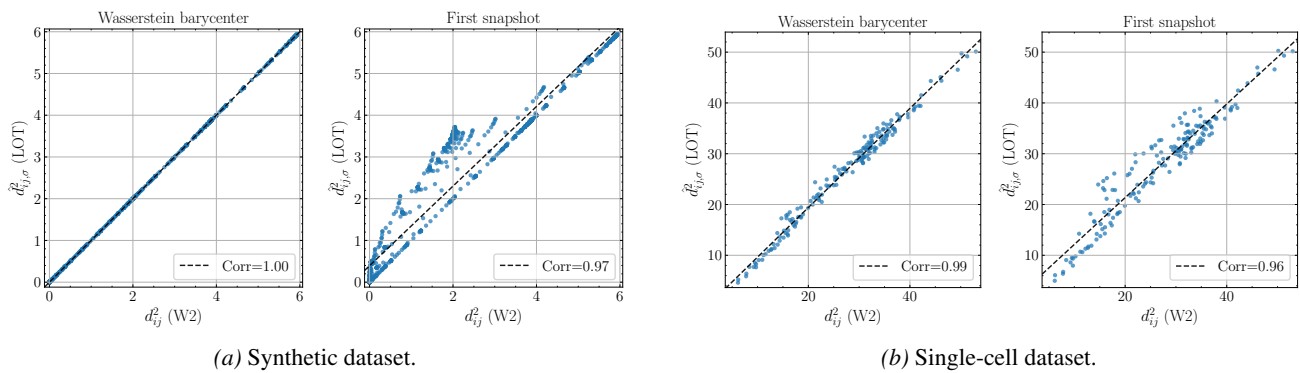

*(a)* Synthetic dataset.  *(b)* Single-cell dataset.

*Figure 8.* We compare the pairwise squared Wasserstein 2 distances ($d_{ij}^2$) with the pairwise squared distances based on LOT embeddings ($\hat{d}_{ij,\sigma}^2$) for two choices of the reference $\sigma$: using the Wasserstein barycenter of the observed snapshots vs. using the first snapshot.

*Table 6.* Runtime summary for the single-cell experiment.

| Component | Total seconds | Avg. seconds/iteration |
|---|---|---|
| Reference computation | 953.765 | 50.198 |
| GP fitting | 158.312 | 8.332 |
| LOT embedding computation | 95.398 | 5.021 |
| Acquisition func. opt. | 4.631 | 0.244 |
| PCA | 3.958 | 0.208 |

## C.5. Alternative reconstruction schemes

Multi-marginal Schrodinger-type methods could in principle be used within our framework for the reconstruction step once snapshots have been acquired with our method. To verify this, we conduct an experiment using MMFM (Rohbeck et al., 2025) to reconstruct a probability path given our acquired snapshots. We report the results in Table 7 and Table 8, where MMFM underperforms our reconstruction framework. We believe this gap is largely due to the strong non-stationarity of the underlying dynamics. Furthermore, unlike in MMSB, our approach can be used to sample multiple plausible probability paths compatible with the observed snapshots. This allows to represent trajectory-level ambiguity explicitly, which is what is leveraged for active learning.

*Table 7.* Comparison with MMFM on the synthetic dataset for the *reconstruction step*. Snapshot timepoints are selected with our method.

| Number of obs. snapshots | Method | Mean $W_2$ | Mean w-$W_2$ |
|---|---|---|---|
| 2 | Ours | 0.906 | 0.924 |
|  | MMFM | 0.913 | 0.951 |
| 12 | Ours | 0.031 | 0.038 |
|  | MMFM | 1.746 | 1.669 |
| 22 | Ours | 0.021 | 0.023 |
|  | MMFM | 1.381 | 1.382 |

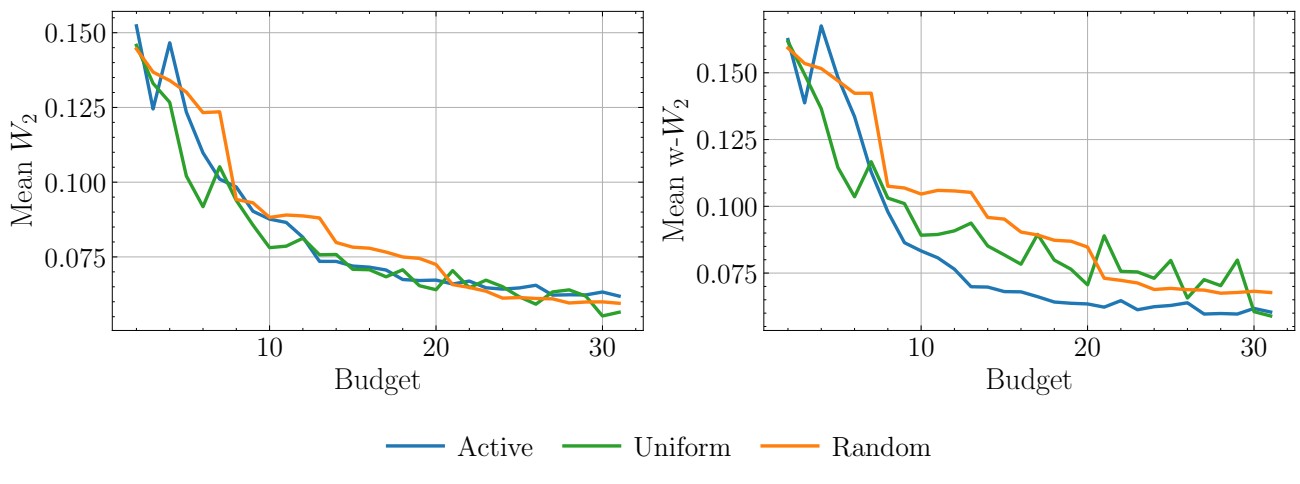

*(a)* Results for the IPUMS-CPS dataset.

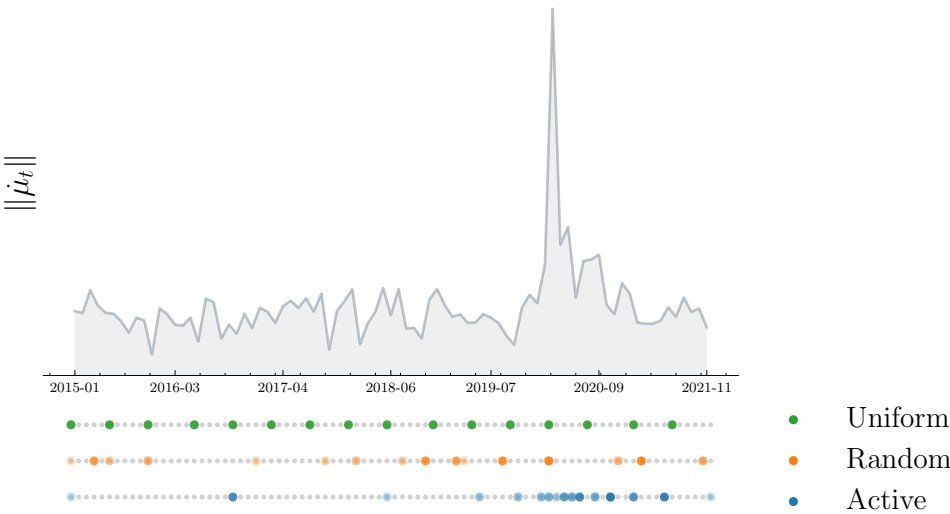

*(b)* Visualization of adaptive timepoint selection for the IPUMS-CPS dataset. **(Top)** The instantaneous metric speed $\|\dot{\mu}_t\|$ of the ground truth trajectory. **(Bottom)** Comparison of selected measurement times. Darker points indicate later acquisitions.

*Figure 9.* Additional results for the IPUMS-CPS dataset.

*Table 8.* Comparison with MMFM on the single-cell dataset for the *reconstruction step*. Snapshot timepoints are selected with our method.

| Number of obs. snapshots | Method | Mean $W_2$ | Mean w-$W_2$ |
|:---:|:---:|:---:|:---:|
| 2 | Ours | 4.631 | 4.763 |
| | MMFM | 4.803 | 4.891 |
| 11 | Ours | 3.048 | 3.165 |
| | MMFM | 4.680 | 4.629 |
| 20 | Ours | 2.906 | 3.048 |
| | MMFM | 4.967 | 4.906 |

