# OpenReview forum: "Active Timepoint Selection for Learning Measure-Valued Trajectories"
_ICML.cc/2026/Conference — ICML 2026 regular_

### Official Review · Reviewer_x1Jh · 2026-02-24

**Soundness:** 4
**Presentation:** 4
**Significance:** 3
**Originality:** 3
**Overall Recommendation:** 5
**Confidence:** 3

**Summary:**

The goal is to infer a trajectory in an *observable* space of probability measures $(\mu_t)_{t \in [0, 1]}$ by querying samples from a finite set $\{ t_1, \ldots, t_N \}$ that is sequentially built by the user.

The authors define a corresponding trajectory in a *latent* space of vectors $(c_t)_{t \in [0, 1]}$, similar to Hidden Markov Models.

- **Mapping from observed to latent space**:  the authors map the probabilities to a vector space (the tangent space), reduce its dimensionality (PCA), and equip it with the standard Euclidean geometry

- **Temporal dynamics in the latent space**: the latent trajectory $(c_t)_{t \in [0, 1]}$ follows a Gaussian Process with time-warping such that the observed trajectory has unit-speed in Wasserstein geometry.

This way, this authors sequentially build $\{ t_1, \ldots, t_N \}$ by minimizing an objective (Eq 13) that quantifies the expected reduction of global uncertainty, naturally measured with the Gaussian Process.

Experiments show that the authors' method is on par or better than naive ways of selecting $\{ t_1, \ldots, t_N \}$.

**Compliance With Llm Reviewing Policy:**

Affirmed.

**Final Justification:**

I have updated my score. The authors have adequately answered my questions and addressed my concerns on whether their method significantly outperforms baselines.

**Key Questions For Authors:**

Q1. Can the authors elaborate on the differences between their approach and the multi-marginal Schrodinger Bridge literature, which also aims to infer a trajectory of probability measures $(\mu_t)_{t \in [0, 1]}$ from finite samples?

Q2. How sensitive are the authors’ results to the choice of the M landmarks (page 4)?

Q3. Can the authors find a setup where their method (“active”) outperforms baselines (“uniform”, “random”) by design?

**Limitations:**

Yes

**Strengths And Weaknesses:**

**Strengths**: To my knowledge, the method is novel, thoughtful, well-motivated and clearly explained. Experiments are introduced pedagogically (Figures 1 and 2) before benchmarks.

**Weaknesses**: In some experiments, the added value of the authors’ method does not seem very significant compared to baselines. Consider Figure 4, which measures performance as a function of the number of points $N$ that discretize time. It is expected that for small or large $N$ (left or right of the graph), all methods are roughly equivalent. Carefully selecting the timepoint matters for moderate values of $N$ (middle of the graph). In this regime, the authors’ method (blue) outperform the baselines but not by a lot.

---

> ### Author Rebuttal · Authors · 2026-03-31
>
> Thank you for your positive and constructive review. We address your comments below.
>
> ---
>
> **All new results (PNG files for figures and tables, e.g. `tab_9_tab_10.png`) are in the following anonymized link: https://anonymous.4open.science/r/rebuttals_33766**
>
> ### [P1] Practical significance in the moderate-budget regime
>
> We agree that the advantage of our method is most visible for moderate values of $N$. This is precisely the regime of practical interest: when $N$ is very small, all methods are under-informed, and when $N$ is very large, simple strategies can densely cover the time interval. The key question is therefore which method uses a *limited measurement budget* most effectively.
>
> In that regime, even a modest improvement is practically meaningful because the dominant cost is acquiring additional snapshots. In applications such as single-cell analysis, sequencing a single new timepoint typically costs on the order of **a few thousand dollars**. Hence reducing the number of measurements needed to achieve a given reconstruction quality translates into substantial experimental savings. For this reason, we view the gains in `Fig. 4` as particularly meaningful despite the moderate absolute gap.
>
>
> ### [P2] Difference from multi-marginal Schrödinger Bridges (MMSB)
>
> The differences between MMSB and our method are the following:
>
> **Different objective.** MMSB methods are *trajectory reconstruction* methods: they infer a dynamics consistent with observed marginals. Our setting is different: we study active acquisition, i.e., which additional timepoint should be measured next under a fixed budget to improve trajectory reconstruction.
>
> **Different output required.** Because our goal is active acquisition, we need a notion of epistemic uncertainty over *possible probability paths*. By contrast, the solution of the MMSB problem (under regularity constraints) is a *single* stochastic process and therefore a *single* induced marginal path. Although the solution may be represented with a stochastic differential equation, it still corresponds to a single probability path, not a distribution over plausible paths.
>
> **Reconstruction.** MMSB-type methods could in principle be used within our framework for the *reconstruction step* once snapshots have been acquired with our method. To verify this, we have conducted a new experiment using MMFM [1] to reconstruct a probability path given our acquired snapshots. We report the results in`tab_9_tab_10.png`, where MMFM underperforms our framework. We believe this gap is largely due to the strong non-stationarity of the underlying dynamics. Finally, unlike in MMSB, our approach can be used to *sample* multiple plausible probability paths compatible with the observed snapshots. This allows to represent trajectory-level ambiguity explicitly, which is what is leveraged for active learning.
>
> **Summary:** MMSB deals with how to reconstruct a *single* path from fixed snapshots, whereas our work addresses how to choose the snapshots in the first place and can model a distribution over probability paths.
>
> [1] Rohbeck, M. et al. Modeling complex system dynamics with flow matching across time and conditions. ICLR 2024
>
> ### [P3] Choice of the landmarks
>
> We choose the landmarks to be a discrete support of the Wasserstein barycenter of the observed snapshots.
>
> **Empirically, this improves performance.** In the ablation study (`Fig. 5`), using a fixed reference given by the first snapshot ("Fixed ref.") leads to consistently worse downstream reconstruction than using the Wasserstein barycenter.
>
> **Distortion analysis.** The LOT representation is a linearization of Wasserstein space around the reference $\sigma$. Its accuracy therefore depends on how well $\sigma$ is centered relative to the observed snapshots. To make this more explicit, we have conducted a new distortion analysis in `fig_8_tab_4_tab_5.png`. For pairs of snapshots $(\mu_i,\mu_j)$, we compare the true squared Wasserstein distance $W_2^2(\mu_i,\mu_j)$ with its LOT approximation $||z_{i,\sigma}-z_{j,\sigma}\||_2^2$. We report the mean absolute error, mean relative error, and correlation, showing that using the Wasserstein barycenter as reference yields substantially lower distortion than using the first snapshot, which explains the improved downstream performance.
>
> ### [P4] Canonical setup
> Our current synthetic setup captures a canonical scenario where our method outperforms baselines since key dynamics are concentrated in a short transition window. To make this point even clearer, we have added a simplified canonical experiment with a single abrupt transition over a very short interval (`fig_11.png`). This setup isolates the key mechanism:  for *Uniform* to succeed, the sampling must be dense enough, and *Random* is unlikely to place a sample in the transition window.
>
> ---
>
> Thank you again for the detailed feedback. We hope our responses have fully addressed your comments, and we would be glad to clarify any remaining points.

---

> > ### Author Rebuttal · Reviewer_x1Jh · 2026-04-04
> >
> > Thank you authors. In essence, all my questions have been answered. I just notice that in Figure 12 in your link, the performance of your method (blue) overlaps substantially with others (green and orange), for low, moderate, or high budgets. So it is not clear to be that there is always a significant enough marginal difference that would translate into saving thousands of dollars. Please correct me if I'm wrong and apologies for the late response.
> >
> > Edit: Thank you authors for the additional figures. I can see that your method and baselines are quite comparable in the high or low budget regimes (Figure 14) and in the low precision regime (Figure 13), which is expected. But outside of these regimes, when we want high precision with a moderate budget, I can see that your method presents statistically significant gains (Figures 13 and 14), and I am willing to believe that they are economically significant too, if "sequencing a single new timepoint typically costs on the order of a few thousand dollars", as you say.
> >
> > I have upgraded my score as you have addressed my concerns. In the final version, I believe it is worth identifying very clearly, in writing, the regime (high precision with moderate budget) in which your method outperforms baselines and the regimes (low precision or high/low budget) when it does not. It is also worth highlighting the financial cost of sequencing a single new time point.

---

> > > ### Author Response · Authors · 2026-04-04
> > >
> > > Thank you for the follow-up, we are glad that all of your questions have been answered. Regarding our newly added `Figure 12`, we agree that the figure alone can make the curves appear visually close, so we added two new figures based on the same results to quantify the gap more directly. We report these figures (`Figures 13 and 14`) at the following link:
> > > https://anonymous.4open.science/r/rebuttals_33766/fig_13_fig_14.png
> > >
> > > **Budget saved.**`Figure 13` reports, for any target reconstruction error, the budget required to reach that target, for both mean $W_2$ and weighted mean $W_2$. In this view, the vertical gap between two curves directly measures the *number of acquisitions saved*. Under this metric, our active method largely requires fewer acquisitions than the baselines. For example, to reach mean $W_2 < 10^{-1}$, it reduces the required budget by about $4$ acquisitions on average relative to the uniform baseline, which shows a non negligible gain.
> > >
> > > **Relative improvement over evaluation times.**  `Figure 14` complements this analysis by reporting the relative improvement of our active method over each baseline. More precisely, for a given seed $s$, budget $b$, and evaluation time $t$, we first compute the relative gain $$g^{(s)}(b,t)=\frac{E_{\mathrm{base}}^{(s)}(b,t)-E_{\mathrm{active}}^{(s)}(b,t)}{E_{\mathrm{base}}^{(s)}(b,t)}$$
> > > where $E_{\mathrm{base}}^{(s)}(b,t)$ and $E_{\mathrm{active}}^{(s)}(b,t)$ denote the reconstruction errors of the baseline and active method, respectively. We then average this quantity over evaluation times,
> > > $\bar g^{(s)}(b)=\frac{1}{|T_{\mathrm{eval}}|}
> > > \sum_{t \in T_{\mathrm{eval}}} g^{(s)}(b,t)$ and finally report the average of these seed-wise quantities across seeds $\mathrm{RI}(b)=\frac{1}{S}\sum_{s=1}^{S} \bar g^{(s)}(b).$
> > > We also report 95\% confidence bands across seeds. In `Figure 14`, the improvement is positive for almost all budgets, except at the extreme low- and high-budget regimes where all methods are either data-starved or close to saturation. Moreover, for almost all budgets, the confidence bands remain above zero, indicating **statistical significance** at the 95\% level.
> > >
> > >
> > > **Summary.** Overall, while `Figure 12` may suggest a modest visual gap at first glance, `Figures 13 and 14` show that the difference is both practically meaningful and statistically significant: our method typically reaches the same reconstruction quality with fewer acquisitions.
> > >
> > > We hope this addresses your question. In case you have any further comments, we would be grateful if you could *edit your previous response* so that we can see them, since, as authors, we cannot view OpenReview official comments posted below this response. Thanks again for the detailed feedback!
> > >
> > > Edit: Thank you for the update, and for taking the time to look at the additional figures. We are glad that they helped clarify the picture.
> > >
> > > We agree with your reading of the results: the clearest advantage of our method is in the high-precision, moderate-budget regime, while the differences are smaller in the low-precision regime and at very low or very high budgets. We will make this much more explicit in the writing. We will also highlight more clearly (with numbers) the practical cost relevance of saving even a few acquisitions, in domains like single-cell analysis.
> > >
> > > Thank you again for the constructive feedback!

---

### Official Review · Reviewer_decN · 2026-03-13

**Soundness:** 2
**Presentation:** 3
**Significance:** 2
**Originality:** 3
**Overall Recommendation:** 4
**Confidence:** 2

**Summary:**

This paper introduces an active learning framework for inferring continuous measure-valued trajectories from sparse, destructive snapshots, addressing critical challenges in high-cost domains like single-cell biology. To navigate the non-Euclidean geometry of the Wasserstein space and the absence of canonical uncertainty quantification, the authors employ Linearized Optimal Transport (LOT) to project distributions into a tangent space, where Multi-Output Gaussian Processes (MOGP) can provide a tractable probabilistic surrogate of the probability path. By incorporating intrinsic time-warping to manage non-stationary dynamics and a geometry-aware acquisition function to minimize integrated posterior variance, the proposed method effectively identifies optimal measurement times, outperforming standard baselines on both synthetic and real-world transcriptomics data.

**Compliance With Llm Reviewing Policy:**

Affirmed.

**Final Justification:**

My concerns have been adequately addressed. I will maintain my positive evaluation.

**Key Questions For Authors:**

Please see the weaknesses.

**Limitations:**

yes

**Strengths And Weaknesses:**

Strengths:
The paper is well-written, offering a conceptually pioneering framework that extends active experimentation to the infinite-dimensional Wasserstein space . Its motivation is compelling, specifically addressing the destructive nature and prohibitive costs of data acquisition in domains like single-cell biology to optimize measurement times under a fixed budget. At the same time, the work is rigorous  with detailed theoretical motivation.

Weaknesses:
1.I have some concerns about the actual utility of this framework particularly regarding the trade-off between computational overhead and empirical gain. The method relies on complex components like Linearized Optimal Transport (LOT) and Gaussian Processes (GP), which introduce significant costs that may not always be justified. As the sensitivity analysis reveals, the performance advantage over uniform sampling is most pronounced in highly non-stationary scenarios with localized branching events ; in more homogeneous or smoother trajectories, the gain diminishes significantly, and the system may even be susceptible to linearization errors . For this framework to be truly impactful in high-stakes fields like single-cell biology, further evidence is needed to demonstrate that the reduction in expensive experimental measurements consistently outweighs the increased complexity and computational demands of the active selection module compared to a denser uniform sampling strategy.

2.To further substantiate the generalizability of the proposed framework, I strongly encourage the authors to include at least one additional real-world experiment outside the domain of single-cell biology. While the paper mentions potential applications in fluid dynamics and macroeconomics , the current evaluation is exclusively focused on biological transcriptomics and a synthetic branching dataset tailored to mimic biological events. Demonstrating the framework's effectiveness on a non-biological measure-valued trajectory (e.g., evolving population distributions in economics or particle density flows in physics) would significantly strengthen the claim that this is a versatile tool for scientific experimentation rather than a domain-specific solution for single-cell trajectory inference.

3. About the theory, my primary concern is the potential failure of Linearized Optimal Transport (LOT) for long-range trajectories, as its $L^2$ approximation of the 2-Wasserstein metric is strictly local . In cases of significant spatial shifts or topological changes, the resulting linearization error will distort the Gaussian Process's uncertainty quantification. This distortion fundamentally undermines the acquisition function, leading to suboptimal measurement selections as the path deviates from the reference measure.

4. Dynamically updating the reference measure $\sigma$ creates a theoretical inconsistency by constantly shifting the tangent space's coordinate system . The paper does not address how the Gaussian Process maintains a stable posterior when the numerical representation of all previous snapshots changes at each iteration. Without proof of consistency under this "coordinate drift," the reliability of the active learning policy remains scientifically unverified.

---

> ### Author Rebuttal · Authors · 2026-03-31
>
> Thank you for your positive and constructive review. We address your comments below.
>
> ---
>
> **All new results (PNG files for figures and tables, e.g. `tab_8.png`) are in the following anonymized link: https://anonymous.4open.science/r/rebuttals_33766**
>
> ### [P1] Computational overhead and experimental cost
>
> We believe the relevant comparison is the cost-benefit trade-off. Our method is motivated by settings such as single-cell sequencing, where the dominant cost is acquiring an additional snapshot (one sequencing costs typically in the order of **a few thousand dollars per time point**), making denser uniform sampling financially impractical.
>
> **Scope of our method.** In that regime, even moderate gains are practically meaningful if they reduce the number of measurements needed. Our claim is therefore that our method is most useful in the setting we target: limited measurement budgets, expensive measurements, and heterogeneous dynamics.
>
> **Empirical evidence.** This is consistent with our results. The advantage over uniform sampling is strongest in *non-stationary settings*, precisely where uniform sampling is most likely to miss brief changing phases (see `Table 1`). As the trajectories become smoother and more homogeneous, the gap naturally decreases.
>
> **Computational cost.** The computational overhead of our method is modest in the intended application regime. In our single-cell experiment, the average runtime is roughly 1 minute per iteration (see `tab_8.png`), which is negligible relative to the cost and turnaround time of performing an additional sequencing experiment.
>
>
>
> ### [P2] Additional real-world dataset
>
> **New experiment.** Following your suggestion, we conducted a new experiment using real-world IPUMS-CPS monthly microdata. Specifically, we use monthly (non-ASEC) U.S. CPS samples from January 2015 to December 2021 and construct a time-indexed sequence of distributions over weekly earnings. For each individual record, we retain observations with `AGE >= 16`, `EARNWT > 0`, and `EARNWEEK > 0`, and represent each month as a weighted empirical measure over `log(EARNWEEK)`, with weights normalized within each month using `EARNWT`.
>
>
> **Results.** As shown in `fig_10.png`, the active strategy performs similarly to the uniform baseline when error is averaged uniformly over calendar time (left panel), and yields a clear improvement under the intrinsic-time (velocity-weighted metric on the right panel). This difference is clear in the bottom panel:  our method preferentially identifies and explores periods where the distribution changes most rapidly. That occurs around the onset of the COVID-19 pandemic, especially in March-April 2020. During this period, the distribution of weekly earnings shifted quickly, because there were many fewer low-earning workers in the data. This shows that our method is a general tool for adaptive experimentation on measure-valued dynamical systems, and is not restricted to biology.
>
> ### [P3] Applicability of LOT
> Because of the character limit, we refer to point **[P1]** in our response to ``Reviewer G4ks`` for a detailed response. To summarize: (i) the quality of the surrogate depends majoritarily on the reference, and using the Wasserstein barycenter substantially reduces distortion relative to a fixed first-snapshot reference (ii) for higher-curvature trajectories, multiple local charts further reduce distortion and improve reconstruction. This highlights an intuitive tractability–expressivity tradeoff: adding charts improves geometric fidelity, but increases the dimensionality of the surrogate and the required number of observations. We view this as a natural and promising direction for future work building on our framework.
>
>
> ### [P4] Consistency of the surrogate
>
> We want to clarify that our method does *not* propagate a posterior across changing tangent charts. Instead, at iteration $n$, we define a new surrogate from the full current dataset $D_n=\\{(t_i,\hat\mu_{t_i})\\}_{i=1}^n$: we recompute the reference $\sigma$, map all observed snapshots into the same current chart, and then fit a GP in that chart. Concretely, if $\phi_n$ denotes the deterministic LOT+PCA embedding associated with $\sigma$, then
> $$
> c\_i^{(n)}=\phi\_n(\hat\mu\_{t_i}), \qquad
> f\_n \sim \mathcal{GP}(0,K\_n), \qquad
> c\_i^{(n)} = f\_n(t_i) + \varepsilon_i,\ \ \varepsilon_i \sim \mathcal N(0,\Sigma\_{\mathrm{obs}}).
> $$
> Thus, the posterior $p(f_n\mid D_n)$ is always defined in one common coordinate system *per iteration*.
>
> **Empirical evidence.** Not updating the reference is in fact worse empirically. Our ablations in `Sec. 5.4` show that fixing the reference degrades performance, whereas recomputing it at each iteration reduces linearization error and yields better reconstructions. We will revise the text to make this point explicit.
>
> ---
>
> Thank you again for the detailed feedback. We hope our responses have fully addressed your comments, and we would be glad to clarify any remaining points.

---

> > ### Author Rebuttal · Reviewer_decN · 2026-04-02
> >
> > Thank you for the detailed feedback.  I will maintain my positive evaluation.

---

> > > ### Author Response · Authors · 2026-04-04
> > >
> > > Thank you for your response, we are glad that your concerns have been fully resolved, and want to thank you for your constructive feedback!

---

### Official Review · Reviewer_G4ks · 2026-03-13

**Soundness:** 3
**Presentation:** 4
**Significance:** 3
**Originality:** 3
**Overall Recommendation:** 4
**Confidence:** 3

**Summary:**

The paper proposes a strategy for probability measure path reconstruction using an active approach to select the time points that will contribute most to improving the quality of the reconstructed trajectory.

**Compliance With Llm Reviewing Policy:**

Affirmed.

**Ethical Review Concerns:**

None.

**Key Questions For Authors:**

View main review.

**Limitations:**

Yes.

**Strengths And Weaknesses:**

The paper is very well written and easy to follow. It does a very good job of motivating the proposed approach and of explaining the challenges in applying standard uncertainty-driven techniques to this specific setting. The motivation and contributions are clearly spelled out. The main contribution is twofold: (1) embedding each snapshot distribution into a common tangent space using Linearized Optimal Transport (LOT), which allows GP modeling and therefore an uncertainty-driven choice of optimal evaluation times; and (2) using a warped multi-output Gaussian Process to deal with non-stationary scenarios.

The pipeline, despite being quite intricate, is sound and clearly presented. Empirically, the method outperforms uniform and random sampling on both synthetic data and a real single-cell reprogramming dataset, with the clearest gains appearing in sparse-budget settings and around sharp, transient high-velocity events. The sensitivity analysis is especially useful, since it shows that the benefit is largest when branching events are localized and shrinks as they become broader.

My main reservation is that the uncertainty is only a surrogate over LOT+PCA latent coordinates, not a native posterior over measures, making it likely very dependent on the applicability or not of LOT and PCA for particular scenarios.  Also, the method relies on a local tangent-space approximation, and the paper itself notes that large distributional shifts can induce linearization errors; same concerns.

On the experimental side, the baselines are relatively weak, since the comparisons are only against random and uniform acquisition. The reviewer wonders whether there is a simpler embedding with active-learning strategy that could serve as a more credible baseline and better justify the complexity of the proposed pipeline.

---

> ### Author Rebuttal · Authors · 2026-03-31
>
> Thank you for your positive and constructive review. We address your comments below.
>
> ---
>
> **All new results (PNG files for figures and tables, e.g. `fig_8_tab_4_tab_5.png`) are in the following anonymized link: https://anonymous.4open.science/r/rebuttals_33766**
>
> ### [P1] Applicability of LOT
> **Tradeoff.** We agree with you that our uncertainty is based on a tractable surrogate defined in LOT+PCA coordinates. This is a deliberate modeling choice, as the core difficulty in our setting is precisely that the output space is infinite-dimensional and non-Euclidean, so there is no canonical GP-like posterior over measures that is both geometry-aware and *computationally tractable*.
>
> **Role of the reference.**  The quality of this surrogate depends on how well LOT  preserves Wasserstein geometry. To reduce the distortion, we choose the reference measure $\sigma$ as the Wasserstein barycenter of the observed snapshots, rather than fixing an arbitrary reference. Our ablations in `Fig. 5` show that this leads to the best downstream performance. To provide more intuition on these gains, we have added a new analysis quantifying the distortion induced by $\sigma$: for pairs $\mu_i,\mu_j$, we compare the true $W_2(\mu_i,\mu_j)^2$ with its LOT approximation $||z_{i,\sigma} - z_{j,\sigma}||_2^2$. We report mean absolute and relative errors, as well as correlations in `fig_8_tab_4_tab_5.png`, and show that using the Wasserstein barycenter as reference yields substantially lower distortion than using the first snapshot as reference.
>
>
>
> **Multiple charts.** Motivated by these results, we investigate whether using *multiple local charts* could improve reconstruction compared to a single reference. In a new experiment (see `fig_9_tab_6_tab_7.png`), we consider rotating anisotropic Gaussians, where $\theta_{\max}$ controls the overall curvature. We show that using two charts (obtained by dividing the interval $[0,1]$ in two and computing a Wasserstein barycenter with observations within each half) *reduces the linearization error* and *improves reconstruction quality* relative to a single-chart LOT representation. While using multiple charts improves performance, it also increases the dimensionality of the representation (roughly linearly with the number of charts) and the number of observations required. Extending this idea into a full active-learning framework would require addressing several additional points: how to define uncertainty consistently across charts and stitch local reconstructions, for example through an atlas-like construction using tools from Riemannian geometry. We view this as a natural and promising direction for future work building on our framework.
>
>
> ### [P2] Baselines
> We have followed your suggestion and have run experiments with these new baselines:
>
> - **Moments embeddings**: we represent any distribution $\mu$ on $\mathbb{R}^d$ by its mean $m_\mu$ and covariance $\Sigma_\mu$, and encode it as $\psi(\mu) = (m_\mu,\operatorname{up tri}(\Sigma_\mu))$ before PCA, where $\operatorname{up tri}$ returns the upper triangular coefficients including the diagonal. At reconstruction time, we decode $\hat\psi$ into $(\hat m,\hat\Sigma)$, project $\hat\Sigma$ to be PSD, and map this to the Gaussian $\mathcal{N}(\hat m,\hat\Sigma)$.
> - **Kernel mean embeddings**:  given a distribution $\mu$, we fix anchors $Z=\\{z\_\ell\\}_{\ell=1}^L$, and, with $k\_\eta(x,z)=\exp(-||x-z||^2/(2\eta^2))$, we define $\psi\_\mu(\ell)=\mathbb{E}\_{x\sim\mu}[k\_\eta(x,z\_\ell)]$.  At reconstruction time, from $\hat\psi$ we recover anchor weights $p$ by optimizing $\min\_{p\in\Delta^L} ||K\_{ZZ}p-\hat\psi ||\_2^2+\lambda ||p||\_2^2$ where $(K\_{ZZ})\_{\ell m}=k\_\eta(z\_\ell,z\_m)$, which yields $\hat\mu=\sum\_{\ell=1}^L p\_\ell \delta\_{z\_\ell}$.
> - **Interval midpoints**: at each acquisition step, we select the candidate time closest to the midpoint of the largest interval between adjacent observed times (both with and without time warping).
> - **Spline uncertainty**: we adapt the acquisition function in [1]. Note that [1] tackles a different problem: selecting time points given measurements from $k$ functions, while we focus on the active learning problem in the space of *distributions*. Nevertheless we adapt it to our problem by using their spline uncertainty acquisition function to our LOT+PCA coefficients.
>
> **Results.** We report the results in `fig_7.png`, where 1) the alternative embeddings yield drastically worse results than with the LOT embeddings, showing the importance of capturing the Wasserstein geometry with OT and 2) our acquisition function, which captures epistemic uncertainty, outperforms the other baselines.
>
> [1] Singh, R. et al., Active learning for sampling in time-series experiments with application to gene expression analysis. ICML 2005
>
> ---
>
> Thank you again for the detailed feedback. We hope our responses have fully addressed your comments, and we would be glad to clarify any remaining points.

---

> > ### Author Rebuttal · Reviewer_G4ks · 2026-04-08
> >
> > I thank the authors for the detailed answer. I will keep my already positive grade.

---

> > > ### Author Response · Authors · 2026-04-08
> > >
> > > Thank you for your feedback, we are glad that your concerns have been fully resolved and appreciate your recommendation to accept the paper.

---

### Official Review · Reviewer_7YbB · 2026-03-13

**Soundness:** 3
**Presentation:** 3
**Significance:** 3
**Originality:** 3
**Overall Recommendation:** 4
**Confidence:** 3

**Summary:**

This paper introduces a novel active learning framework designed to optimize the timing of the data collection process in fields where sampling is "destructive" and expensive, for example, single-cell biology. They proposed Linearized optimal transport, which enables mapping complex distributional snapshots into a simpler tangent space. By applying GP to these tangent vectors, the model provides a way to quantify epistemic uncertainty—identifying exactly where the model is "unsure" about the trajectory. Experimental results also support the proposed hypothesis.

**Compliance With Llm Reviewing Policy:**

Affirmed.

**Key Questions For Authors:**

Uniform and random baselines seem overly simplistic; it would be valuable to compare the proposed method against stronger, more sophisticated alternatives. In particular, the paper argues that probability measures evolve on a Wasserstein space rather than a Euclidean one, implying that standard Gaussian Process models are inadequate, but this point would be much more convincing if it were supported by targeted empirical comparisons that directly illustrate this failure mode.

Computational complexity could be prohibitive for very large datasets or real-time active selection, right?

The proposed method relies on the Linearized Approximation of Optimal Transport (LOT). However, I wonder how the method performs when the trajectory has very high curvature.

**Limitations:**

Whether the proposed framework scales effectively to much higher-dimensional latent spaces without becoming computationally challenging remains unclear.

One of the major concerns is the baseline. Current baselines are too naive.

**Strengths And Weaknesses:**

Strengths:

The paper identifies and formalizes a significant gap in active time slot selection for measurement in path measure space, extending it to the space of measures. This is highly relevant for destructive sampling fields like single-cell biology, where traditional Euclidean active learning methods are inapplicable.

The method demonstrates clear advantages over uniform and random baselines, particularly in low-budget regimes. Both the qualitative and quantitative analysis specifically shows the model’s strength in capturing sharp, transient dynamics.

The use of Linearized Optimal Transport (LOT) is a nice way to handle the problem. By mapping distributions to a tangent space, the authors can leverage established probabilistic models like Gaussian Processes.

The writing is good and easy to follow, and the motivation is strong.


Weakness:

The experimental supports are too narrow. Not sure about the scalability in a large dimensional setting.

The baselines are weak.

typo: missing a reference in line 117.

---

> ### Author Rebuttal · Authors · 2026-03-31
>
> Thank you for your positive and constructive review. We address your comments below.
>
> ---
> **All new results (PNG files for figures and tables, e.g. `fig_6.png`) are in the following anonymized link: https://anonymous.4open.science/r/rebuttals_33766**
>
> ### [P1] Failure of naive Euclidean GP interpolation
>
> We agree that this point is best supported with an experiment. We consider a new setup with a 1D Gaussian trajectory whose mean moves over time, see `fig_6.png`. We observe the distributions at $t=0, 0.5, 1$, represent each of them by its density function computed on a fixed grid, and fit a standard Gaussian process directly on these density vectors. We then predict the distribution at $t=0.25$. In `fig_6.png`, the naive Euclidean interpolation leads to mass splitting rather than the correct transport. This illustrates the key problem: naive GP regression in density space ignores the Wasserstein geometry of distributions, which is what motivates our LOT-based construction.
>
> ### [P2] Baselines
> We have followed your suggestion and have run experiments with these new baselines:
>
> - **Moments embeddings**: we represent any distribution $\mu$ on $\mathbb{R}^d$ by its mean $m_\mu$ and covariance $\Sigma_\mu$, and encode it as $\psi(\mu) = (m_\mu,\operatorname{up tri}(\Sigma_\mu))$ before PCA, where $\operatorname{up tri}$ returns the upper triangular coefficients including the diagonal. At reconstruction time, we decode $\hat\psi$ into $(\hat m,\hat\Sigma)$, project $\hat\Sigma$ to be PSD, and map this to the Gaussian $\mathcal{N}(\hat m,\hat\Sigma)$.
> - **Kernel mean embeddings**:  given a distribution $\mu$, we fix anchors $Z=\\{z\_\ell\\}_{\ell=1}^L$, and, with $k\_\eta(x,z)=\exp(-||x-z||^2/(2\eta^2))$, we define $\psi\_\mu(\ell)=\mathbb{E}\_{x\sim\mu}[k\_\eta(x,z\_\ell)]$.  At reconstruction time, from $\hat\psi$ we recover anchor weights $p$ by optimizing $\min\_{p\in\Delta^L} ||K\_{ZZ}p-\hat\psi ||\_2^2+\lambda ||p||\_2^2$ where $(K\_{ZZ})\_{\ell m}=k\_\eta(z\_\ell,z\_m)$, which yields $\hat\mu=\sum\_{\ell=1}^L p\_\ell \delta\_{z\_\ell}$.
> - **Interval midpoints**: at each acquisition step, we select the candidate time closest to the midpoint of the largest interval between adjacent observed times (both with and without time warping).
> - **Spline uncertainty**: we adapt the acquisition function in [1]. Note that [1] tackles a different problem: selecting time points given measurements from $k$ functions, while we focus on the active learning problem in the space of *distributions*. Nevertheless we adapt it to our problem by using their spline uncertainty acquisition function to our LOT+PCA coefficients.
>
> **Results.** We report the results in `fig_7.png`, where 1) the alternative embeddings yield drastically worse results than with the LOT embeddings, showing the importance of capturing the Wasserstein geometry with OT and 2) our acquisition function, which captures epistemic uncertainty, outperforms the other baselines.
>
>
> [1] Singh, R. et al., Active learning for sampling in time-series experiments with application to gene expression analysis. ICML 2005
>
> ### [P3] Scalability
>
> **Computational cost.**  In our intended application regime, the computational overhead is modest (see `tab_8.png`): in the single-cell experiment, one active-learning iteration takes about **1 minute**. In settings such as active sequencing, this overhead is negligible relative to the cost and turnaround time of performing an additional sequencing experiment.
>
> **Scalability to larger datasets.** The dominant cost is the OT computations. We agree that, for *very large datasets*, exact OT can become a bottleneck. We want to clarify that this is a limitation of the current implementation rather than of the overall framework. In particular, our method can be combined with standard scalable OT approximations, such as  Sinkhorn solvers with entropic regularization or low-rank OT methods. We will revise the manuscript to explicitly discuss the computational complexity and highlight these practical scaling options.
>
> ### [P4] Linearization and curvature
> Because of the character limit, we refer to point **[P1]** in our response to ``Reviewer G4ks`` for a detailed response. To summarize: (i) the quality of the surrogate depends majoritarily on the reference, and using the Wasserstein barycenter substantially reduces distortion relative to a fixed snapshot reference (ii) for higher-curvature trajectories, multiple local charts further reduce distortion and improve reconstruction. This highlights a tractability–expressivity tradeoff: adding charts improves geometric fidelity, but increases the dimensionality of the surrogate and the number of observations required. We view this multi-chart idea as a natural and promising direction for future work building on our framework.
>
> ---
> Thank you again for the detailed feedback. We hope our responses have fully addressed your comments, and we would be glad to clarify any remaining points.

---

> > ### Author Rebuttal · Reviewer_7YbB · 2026-04-01
> >
> > I thank the Authors for their detailed responses. The Authors have adequately addressed some of my concerns. However, scalability remains a major bottleneck. Although the authors suggested a few workarounds, I am not fully convinced. For example, Sinkhorn solvers with entropic regularization still suffer in large dimensions. Thus, I am not sure if combining the proposed method with Sinkhorn solvers with entropic regularization effectively solves the scalability issues. Thus, I keep my score, and I vote for Acceptance.

---

> > > ### Author Response · Authors · 2026-04-04
> > >
> > > Thank you for the follow-up, we are glad that most of your concerns have been addressed and that you recommend acceptance.
> > >
> > > We agree that scalability is an important point. To be clear, we do *not* claim that standard Sinkhorn solvers fully resolves scalability in the *most extreme regime*, e.g. *million-point* snapshots. We agree that, in *such settings*, large-scale OT can be challenging both computationally and in memory.
> > >
> > >
> > > **Practical regime.** Our point is the following: this limitation does **not** prevent the method from being practical in the application regime that the paper targets. The intended regime in our paper is active acquisition, where obtaining a new snapshot is itself expensive and slow. For example, in active sequencing, the dominant cost is the biological experiment and sequencing turnaround, not the acquisition rule. In our current single-cell experiment, the runtime of one active-learning iteration is about **1 minute**, which is modest relative to the cost and latency of performing an additional sequencing measurement.
> > >
> > >
> > > **New experiment.** To demonstrate a path to scaling for hundreds of thousands of points, we have now run a substantially larger-scale experiment. Specifically, we scale the synthetic setup to **100,000 samples per snapshot**, with $M=1000$ reference landmarks for the reference measure $\sigma$. We use the OTT-JAX library, and run the experiments on an RTX 3080 GPU. We use scalable OT solvers in the components where they matter most:
> > >
> > > * for the reference computation, we use the Sinkhorn solver, together with a memory-efficient implementation for the free-support barycenter computation that avoids materializing large 3D tensors in memory. We use the same Sinkhorn solver for the LOT embedding computations.
> > > * for the pairwise OT problems used in the time-warping step, we use low-rank Sinkhorn with rank $256$.
> > >
> > > The runtime breakdown is in the following table:
> > >
> > > | Component                         |   Total time | Avg. per acquisition iteration |
> > > | --------------------------------- | -----------: | -----------------------------: |
> > > | Reference computation             | 1h 21m 50.0s |                       3m 53.8s |
> > > | Warp construction                 | 1h 13m 47.2s |                       3m 30.8s |
> > > | LOT embedding computation         |    29m 23.8s |                       1m 24.0s |
> > > | GP fitting                        |       46.03s |                          2.19s |
> > > | Acquisition function optimization |        3.27s |                          0.16s |
> > > | PCA                               |        0.80s |                          0.04s |
> > >
> > > This corresponds to an average of roughly 8.9 minutes per acquisition iteration in the 100k-point setting. The resulting interpolation quality (see https://anonymous.4open.science/r/rebuttals_33766/fig_12.png) remains aligned with those observed in the smaller-scale setup.
> > >
> > > We stress that we do not present this as "solving" OT scalability in general. Rather, we view it as a piece of evidence that, with scalable OT solvers, the framework remains practical beyond the scale of the current experiments (which already have thousands of points per snapshot for the single-cell dataset, and tens of thousands of points per snapshot for the newly added IPUMS-CPS dataset described in our response to *Reviewer decN*).
> > >
> > > **Summary.** We fully agree that **extreme-scale** regimes (million-point snapshots) can be challenging, and we do not want to overstate this. However, this limitation does not undermine the main claim of the paper, because the paper targets a different regime: expensive sequential acquisition problems where snapshots are costly to acquire, budgets are small, and the decision rule is run between measurements rather than under hard real-time constraints. In that regime, the empirical runtimes we report are practical.
> > >
> > > **Updates to the manuscript.** We agree that this should be stated more explicitly in the manuscript. We will revise the paper accordingly in three ways:
> > >
> > > (1) we will clarify that the intended regime is active acquisition with expensive snapshots, not real-time selection
> > >
> > > (2) we will add the new 100k-sample-per-snapshot experiment above to show a concrete scaling path to hundreds of thousands of points
> > >
> > > (3) we will explicitly state that **million-point scaling remains an open challenge** for the current implementation, due to both compute and memory constraints, rather than implying that standard Sinkhorn alone completely resolves it, i.e. `` "The main computational cost of our framework comes from the OT subroutines. In the intended application regime of this paper, i.e. active acquisition settings with expensive snapshots and up to roughly
> > > 10^5 samples per snapshot, this overhead is practical. We do not claim that million-point snapshots are fully solved: in that regime, both OT computation and memory become major constraints, and additional scalable OT approximations would be required." ``

---

### Decision · Program_Chairs · 2026-04-30

**Decision:**

Accept (regular)

**Comment:**

This manuscript concerns an application of Bayesian experimental design for optimizing measurement times during time-evolving experiments e.g. in cell biology.

The reception to this work during the initial review stage was positive, with reviewers praising the clarity of the presentation and the approach to modeling and problem formulation. Some reviewers, however, expressed concerns regarding the scope and analysis of the empirical study. That said, following the discussion period, there was universal agreement among the reviewers that the paper was a solid contribution to the conference, and there was universal recommendation of acceptance.

Further, I believe the novelty of the work and problem setting, combined with a technically strong/solid approach, would make this an excellent contribution to the conference.

I encourage the authors to take the feedback from the reviewers into account while revising the manuscript.